# ZBTB12 is a molecular barrier to dedifferentiation in human pluripotent stem cells

Dasol Han [1,6], Guojing Liu[1,2,6], Yujeong Oh [3,6], Seyoun Oh[3], Seungbok Yang[3], Lori Mandjikian[1], Neha Rani[1,4], Maria C. Almeida [1,5], Kenneth S. Kosik [1] ✉ & Jiwon Jang [3] ✉

Development is generally viewed as one-way traffic of cell state transition from primitive to developmentally advanced states. However, molecular mechanisms that ensure the unidirectional transition of cell fates remain largely unknown. Through exact transcription start site mapping, we report an evolutionarily conserved BTB domain-containing zinc finger protein, ZBTB12, as a molecular barrier for dedifferentiation of human pluripotent stem cells (hPSCs). Single-cell RNA sequencing reveals that ZBTB12 is essential for three germ layer differentiation by blocking hPSC dedifferentiation. Mechanistically, ZBTB12 fine-tunes the expression of human endogenous retrovirus H (HERVH), a primate-specific retrotransposon, and targets specific transcripts that utilize HERVH as a regulatory element. In particular, the downregulation of HERVH-overlapping long non-coding RNAs (lncRNAs) by ZBTB12 is necessary for a successful exit from a pluripotent state and lineage derivation. Overall, we identify ZBTB12 as a molecular barrier that safeguards the unidirectional transition of metastable stem cell fates toward developmentally advanced states.

Stem cell states are recently recognized to be highly metastable[1,2]. Pluripotent stem cells (PSCs) provide a well-defined model system to understand molecular principles of dynamic stem cell states. PSCs lie in a wide spectrum of pluripotent states, ranging from naïve to primed states. Pre- and post-implantation epiblasts in vivo respectively represent naïve and primed pluripotent states[1,2]. Recent studies have further shown an intermediate state called formative pluripotency that exists in a developmental continuum between the naïve and primed states and represents a critical period for lineage capacitation[3–8]. Naïve PSCs show resistance to differentiation and the naïve through formative to primed transition that occurs in peri-implantation development is essential for pluripotency exit and three germ layer differentiation[6,9–11]. Furthermore, mouse PSCs cultured with serum exhibit substantial intra-population heterogeneity with a mixture of naïve- and primed-like cells, which represents a dynamic equilibrium of cells transitioning between distinct metastable states[12,13]. Although human embryonic stem cells (hESCs) are established from inner cell mass of pre-implantation blastocysts, they are more likely to be in a primed pluripotent state[1,2]. Recently, a number of studies have introduced different combinations of chemical inhibitors to drive hESCs to a naïve-like pluripotent state[1]. Like serum-maintained mouse PSCs, the presence of naïve-like cells was also reported in a conventional primed hESC population[14] although the percentage of naïve-like cells is much lower in hESCs than in mESCs. Given the high degree of flexibility in stem cell states, it is plausible that as cells exit from a primed pluripotent state, hESCs would confront either differentiation into three germ layers or dedifferentiation toward a naïve-like state. The concept of embryonic dedifferentiation is supported by recent studies about

[1]Neuroscience Research Institute, Department of Molecular, Cellular, and Developmental Biology, University of California, Santa Barbara, CA, USA. [2]Novogene Co., Ltd, Beijing, China. [3]Department of Life Sciences, Pohang University of Science and Technology (POSTECH), Pohang, Korea. [4]Department of Biological Sciences & Bioengineering, Indian Institute of Technology, Kanpur, India. [5]Federal University of ABC, Center for Natural and Human Sciences São Bernardo do Campo, Santo André, Brazil. [6]These authors contributed equally: Dasol Han, Guojing Liu, Yujeong Oh. ✉e-mail: kenneth.kosik@lifesci.ucsb.edu; jiwonjang@postech.ac.kr

neural crest cell development, in which natural dedifferentiation occurs to drive neural crest cell specification from neuroectoderm[15]. However, it remains unknown how dedifferentiation is blocked once stem cells undergo differentiation.

Endogenous retroviruses (ERVs) are genetic remnants of retro-viral infection during evolution and encompass about 8% of the human genome[16,17]. Increasing evidence suggests that ERVs have significantly contributed to rewiring regulatory networks by creating new tran-scription factor (TF) binding sites[18,19]. Given the species-specific ERV integration events and high ERV expression in the early development, it has been proposed that ERVs play key roles in dictating species-specific developmental programs[20,21]. In particular, human endogen-ous retrovirus H (HERVH) is a primate-specific ERV family and exhibits high expression in human blastocyst stages[22]. HERVH-associated long terminal repeat 7 (LTR7) harbors binding sites for several key plur-ipotency factors such as NANOG, OCT4 (also known as POU5F1), KLF4, and TFCP2L1 (also known as LBP9), and thereby acts as strong pro-moters and enhancers in hPSCs[23]. Recently, it was reported that ele-vated expression of HERVs could be a key feature of naïve-like pluripotent states[14,24,25], raising the possibility that primate-specific HERVs could serve as a molecular rheostat for dynamic human plur-ipotent states. Likewise, retrotransposons including ERVs and LINEs regulate the cell fate transition between two-cell embryos and blas-tocysts in mice[26,27]. Beyond embryonic development, it has been increasingly reported that cellular dedifferentiation occurs in various types of cancers with elevated HERV expression[28]. Therefore, it would be crucial to investigate precise roles and regulation of HERVs in dynamic cell state transition to understand human diseases.

Previous studies have been focusing on TFs that exhibit dynamic expression changes with strong and specific expression in stem cells compared to differentiated cells. Such studies resulted in identifi-cation of core pluripotency factors such as OCT4, NANOG, and SOX2[29–31]. However, in this study we employ a different approach that is not dependent on expression changes of TFs. Rather, we precisely map transcription start sites (TSSs) of transcripts that show differ-ential expression patterns during stem cell differentiation and then identify potential transcriptional regulators by TF motif analysis on promoter sequences. This approach enables us to discover a new pluripotency regulator with broad expression in multiple cell types and tissues. ZBTB12, a BTB (BR-C, ttk and bab) domain-containing zinc finger protein, is expressed in the nucleus of hESCs and balance self-renewal and differentiation. ZBTB12 depletion skews hESCs toward self-renewal and delays the exit from a pluripotent state. Single cell transcriptomic analysis reveals that upon differentiation ZBTB12-depleted hESCs aberrantly transit backward to a naïve-like state and thus fail to differentiate into three germ layers. This sug-gests that ZBTB12 serves as a barrier of dedifferentiation. As a downstream mechanism, we find that ZBTB12 transcriptionally reg-ulates HERVH expression. In contrast to KRAB domain-containing ZFPs (KRAB-ZFPs) that have co-evolved with target ERVs[17], ZBTB12 is an evolutionarily conserved gene in vertebrates, yet it has acquired a regulatory function on specific transcripts that make use of LTR7/HERVH promoters during primate evolution. These findings provide an interesting example of host cells utilizing an old regulator to deal with newly evolved genetic elements. A growing body of recent evi-dence suggests that the genomic integration of HERVH with its strong promoter activity substantially contributed to the evolu-tionary emergence of primate-specific long non-coding RNAs (lncRNAs)[32]. Consistent with this idea, ZBTB12 suppresses a number of HERVH-overlapping lncRNAs including LINC-ROR and ESRG. ZBTB12-mediated shutdown of the lncRNAs drives the exit from a pluripotent state and safeguards three germ layer differentiation. Overall, we have identified ZBTB12 as a molecular barrier that restricts dynamic stem cell states and safeguards stem cell differentiation.

## Results

### Identification of ZBTB12 as a key pluripotency regulator by nanoCAGE-seq

To identify TFs that regulate hESC differentiation, we established an hESC-derived neural differentiation model[33] and performed nanoCAGE sequencing from cells collected at Day 0, Day 5, Day 15, and Day 23 with two replicates at each time point (Supplementary Fig. 1a, b). The nanoCAGE method captures the 5′ends of transcripts and thereby enables genome-wide mapping of transcription start sites (TSSs) as well as transcriptome profiling[34]. The reproducibility of the nanoCAGE method was validated by the principal component analysis and the correlation analysis for all the samples (Supplementary Fig. 1c, d). Community detection analysis based on 3712 differentially expressed genes (DEGs) resulted in three modules that represent three sequential transition stages during neural differentiation (Supplementary Fig.1e–g). Given the high correlation between replicates (Supplemen-tary Fig. 1d), we merged the sequencing data of replicates for TSS peaks identification with HOMER (http://homer.ucsd.edu/homer/). With stringent criteria that an individual peak is supported by at least 30 unique reads with high confidence, 11,019–12,503 peaks were identified in each sample (Supplementary Data 1). Overall, about 70% of the peaks were annotated as TSS peaks due to their location near annotated TSSs (<1 kb from nearest TSS). Other peaks were far from annotated TSSs, representing 3′UTRs, intergenic regions, intronic regions, and exonic regions (Supplementary Fig. 1h, i).

Given that our data accurately identified the TSS position of each transcript, we extracted 300 bp promoter sequences (250 bp upstream and 50 bp downstream of TSS) of stage-specific transcripts. TF motif prediction with HOMER uncovered dynamic changes of TFs throughout sequential cell fate transitions during neural differentia-tion of hESCs (Supplementary Table 1). The number of stage-specific peaks and predicted TFs suggest that widespread transcriptional changes occur during early stages of pluripotency exit (Supplementary Fig. 1j, k). Therefore, we focused on TFs with enriched motifs on Day 0. These TFs include well-known pluripotency factors, such as OCT4, NANOG, SOX2, MYC, and KLF5, and provide confidence in our results (Fig. 1a). Of particular interest was ZBTB12, a zinc finger protein that contains a BTB domain and four zinc finger C2H2 domains, because its functions were unknown. Similar to the core pluripotency factor OCT4, endogenous ZBTB12 proteins were detected in the nuclei of hESCs by immunofluorescence staining (Fig. 1b, Supplementary Fig. 2a). Tagged exogenous ZBTB12 proteins also showed nuclear localization (Supplementary Fig. 2b). To investigate the role of ZBTB12 in hESCs, we downregulated ZBTB12 with two short hairpin RNAs (shRNAs) (Supplementary Fig. 2c, d). ZBTB12 knockdown (KD) in hESCs did not affect pluripotent state maintenance for over ten pas-sages (Supplementary Fig. 2e). Founder cells in hESC colonies have a repopulating capacity when seeded at a clonal density[35,36]. Strikingly, ZBTB12 KD boosted the colony-initiating capacity with increased number of alkaline phosphatase (AP)-positive colonies (Fig. 1c, Sup-plementary Fig. 2f). Furthermore, exogenous expression of ZBTB12 reduced the colony-initiating capacity (Fig. 1d, Supplementary Fig. 2g, h). These findings lead to an interesting concept that undifferentiated hESCs express ZBTB12 in order to restrict their self-renewal ability.

In response to differentiation cues, self-renewing hESCs undergo rapid dissolution of the core pluripotency network to enable plur-ipotency exit and lineage induction[37]. Given the antagonistic role of ZBTB12 in stem cell self-renewal, we hypothesized that ZBTB12 KD could shift the balance of a pluripotent state toward self-renewal and away from differentiation. To test this hypothesis, hESCs were spon-taneously induced to exit from a pluripotent state by FGF2 and TGFβ deprivation. Control hESCs showed downregulation of core plur-ipotency factors, implicating successful exit from a pluripotent state (Fig. 1e). However, ZBTB12 KD strongly impeded pluripotency exit (Fig. 1e, f, Supplementary Fig. 2i, j). In accordance with retained core

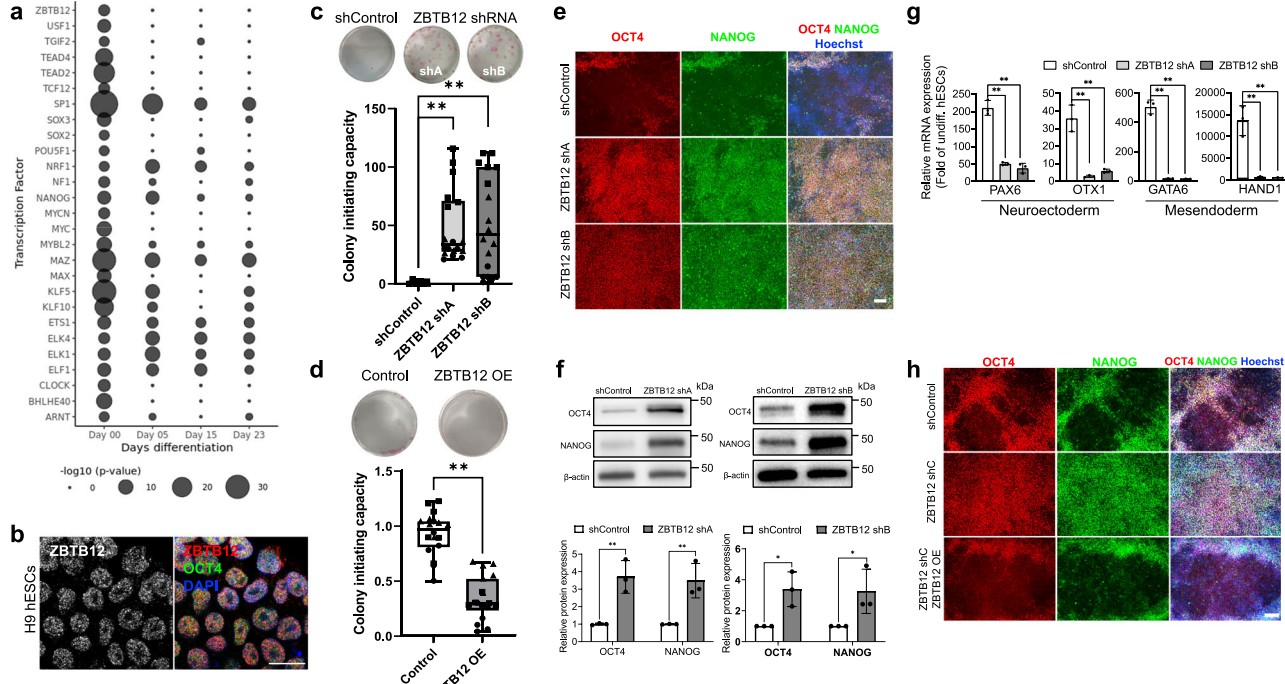

**Fig. 1 | nanoCAGE identifies a new pluripotency regulator ZBTB12. a** Bubble plot showing Day 0-enriched transcription factors during neuronal differentiation process predicted with HOMER based on promoter sequences (300 bp around TSS) of stage-specific transcripts. Twenty-seven transcription factors with $p < 0.01$ and high expression (Cortecon normalized > 500 average read counts at day 0 from Cortecon database, GSE56796, https://cortecon.neuralsci.org) were selected and shown. Size of the circle represents the $-\log_{10} p$ value of transcription factors. *P* value was corrected by the Benjamini–Hochberg method. **b** Representative images from three independent immunofluorescence assays for ZBTB12 and OCT4 in H9 hESCs. Scale bar, 20 μm. **c, d** Colony forming assay with H1 hESCs transduced with lentiviral vectors expressing ZBTB12 shRNAs (**c**) or H9 hESCs transduced with lentiviral vectors expressing ZBTB12 (**d**). Representative alkaline phosphatase (AP)-stained wells are shown. The number (colony initiating capacity) of colonies is quantified. Data are represented as box-plot with median and min to max with all data point. ($n = 12$ for **c** and $n = 18$ for **d** from three independent experiments represented by different point shapes). Center line, median; box limits, upper and lower quartiles; whiskers, max and min; points, all data points. Student's *t* test (two tailed, unpaired) \*\**p* < 0.01. **e** Representative images from three independent immunofluorescence assays for OCT4 and NANOG in differentiated (day 5) H9 cells expressing either shControl or ZBTB12 shRNAs. Scale bar, 100 μm. **f** Western blots of stem cell markers (top, OCT4 and NANOG) and quantitation (bottom) in H9 hESCs upon FGF2 and TGFβ deprivation for 5 days. Error bars represent mean ± sd ($n = 3$ Student's *t* test (Holm-Sidak's multiple unpaired) \**p* < 0.05 and \*\**p* < 0.01 compared to shControl. **g** qPCR analysis of neuroectoderm (PAX6 and OTX1) and mesendoderm (GATA6 and HAND1) markers in differentiated H9 cells (Day 8) expressing either shControl or ZBTB12 shRNAs. Error bars represent mean ± sd ($n = 3$). Student's *t* test (two tailed, unpaired) \*\**p* < 0.01 compared to shControl. **h** Representative images from three independent immunofluorescence assay for OCT4 and NANOG in differentiated (day 5) H9 cells expressing ZBTB12 shC (3'UTR targeting shRNA) with or without ZBTB12 overexpression. Scale bar, 100 μm. All data points and the exact *p*-values can be found in the Source Data file.

pluripotency gene expression, the activation of neuroectodermal and mesendodermal markers was dramatically perturbed by ZBTB12 KD (Fig. 1g, Supplementary Fig. 2k). These results were further confirmed in three germ layer differentiation via embryoid body formation and neuroectoderm-directed differentiation by dual SMAD inhibition (Supplementary Fig. 2l, m). Inducible KD experiments revealed that the impeded pluripotency exit by ZBTB12 KD is a reversible phenotype (Supplementary Fig. 2n–p). In the opposite experiment, exogenous expression of ZBTB12 further promoted pluripotency exit and lineage derivation (Supplementary Fig. 2q, r), and rescued the ZBTB12 KD-driven defect of pluripotency exit (Fig. 1h, Supplementary Fig. 2s, t). Overall, these data suggest that ZBTB12 is a key regulator to conserve the balance between stem cell self-renewal and differentiation.

## Necessity of ZBTB12 for pluripotency exit revealed by single-cell RNA sequencing

Given the dynamic nature of stem cell differentiation, we applied droplet-based single-cell RNA sequencing (scRNA-seq) to clarify the role of ZBTB12 in pluripotency exit. H9 hESCs expressing either control or ZBTB12-targeting shRNA were differentiated for 8 days by FGF2 and TGFβ withdrawal, followed by scRNA-seq. A dataset of 8,618 cells (Supplementary Fig. 3a–f, Methods) partitioned into seven distinct uniform manifold approximation and projection (UMAP) clusters

(Fig. 2a, b)[38]. Based on the representative marker genes enriched in each cluster (Fig. 2c, d), three major cell states were annotated (Fig. 2e). Clusters 0, 1, and 2 were associated with high expression of pluripotent stem cell markers (OCT4, NANOG, SOX2, and ESRG). Clusters 3 and 6 were annotated as the neuroectodermal cells with a robust expression of OTX1, PAX6 and ZBTB17. Clusters 4 and 5 represented the mesendodermal lineages with GATA6 and HAND1 expression (Fig. 2c–e). Undifferentiated hESCs (Day 0 shCtrl and Day 0 shZBTB12) were spread across pluripotent stem cell clusters (Fig. 2b, e, f). Although ZBTB12 KD did not alter the cell distribution on the UMAP space in undifferentiated cells, we explored DEGs since ZBTB12 KD cells showed a significantly higher colony-initiating ability and impaired differentiation potential (Fig. 1c, e–g). ZBTB12 KD showed mild transcriptional alteration in protein coding genes with 64 upregulated (log2FC > 0.25) and 130 downregulated (log2FC < −0.25) DEGs (Supplementary Data 2). Downregulated DEGs include several developmental genes related to FGF, BMP, and WNT signaling pathways (FGFR1, ID1, ID2, ID4, SFRP1) (Supplementary Fig. 3g). Gene ontology (GO)[39] analysis using metascape revealed that the down-regulated DEGs are associated with GO terms related to stem cell differentiation (GO:0030855-epithelial cell differentiation, GO:0048729-tissue morphogenesis, GO:0060322 head development) (Supplementary Fig. 3h). GO analysis with up-regulated DEGs showed cholesterol metabolism

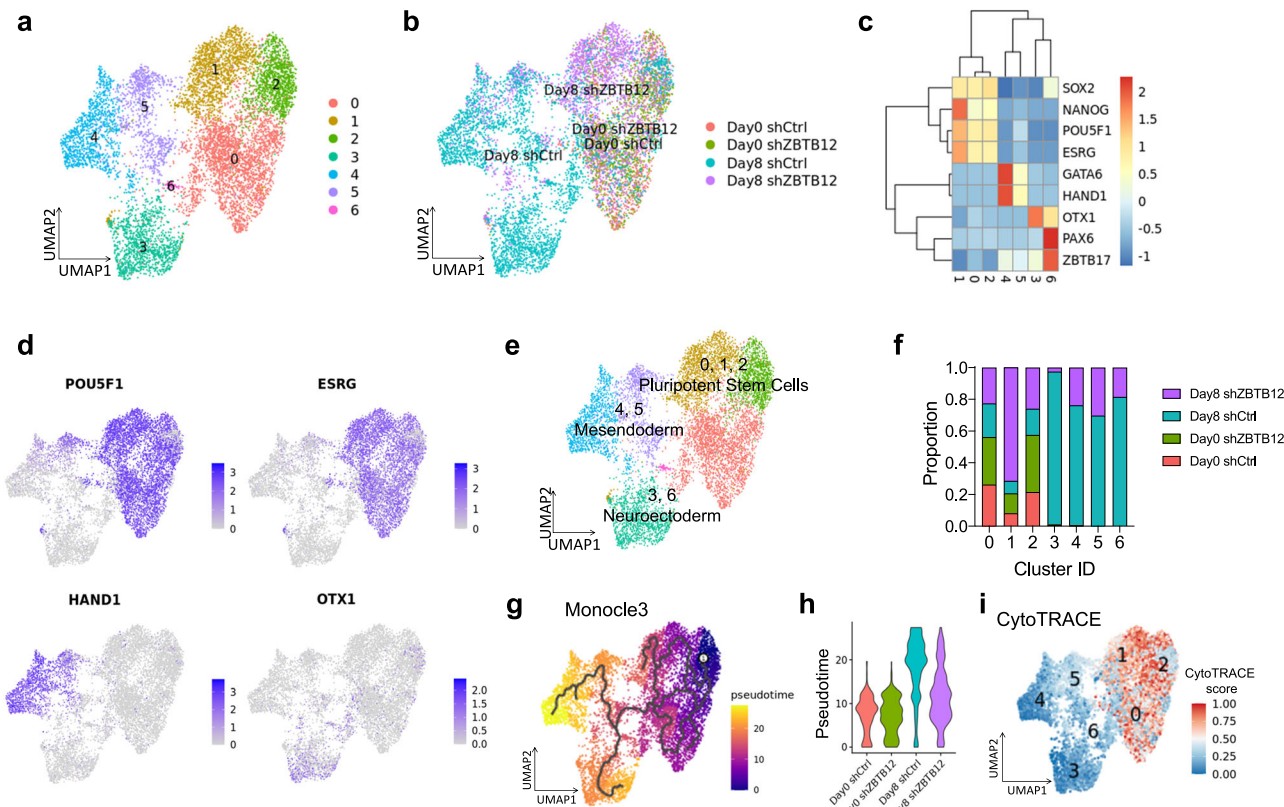

**Fig. 2 | Single cell RNA sequencing reveals the necessity of ZBTB12 for exit from pluripotency. a, b** UMAP plots of 8,618 cells after QC (Supplementary Fig. 3a–f), normalization, dimensionality reduction and clustering, colored by cluster identities (**a**) or by experimental groups (**b**). **c** Heatmap of representative marker gene expression of pluripotency (SOX2, NANOG, POU5F1 and ESRG), mesendoderm (GATA6 and HAND1) and neuroectoderm (OTX1, PAX6, and ZBTB17). **d** Feature plots of gene expression associated with pluripotency (POU5F1 and ESRG), mesendoderm (HAND1) and neuroectoderm (OTX1). **e** Cell state annotation using defined marker gene expression. **f** Cellular compositions of each cluster. Data points can be found in the Source Data file. **g** Cellular trajectory reconstruction using Monocle 3 on UMAP plot, color-mapped by pseudotime. **h** Violin plots showing the distribution of pseudotime (from Fig. 2g) within each group. **i** Cellular trajectory reconstruction using CytoTRACE, color-mapped by CytoTRACE score (higher score indicates less differentiated state).

with the most significant *p* value (Supplementary Fig. 3h). Activated lipid metabolism was recently reported to be a key feature of developmentally earlier stage of pluripotency[8]. Overall, these results are consistent with the elevated self-renewal ability and impaired differentiation potential by ZBTB12 suppression (Fig. 1c, e–g). Upon differentiation for 8 days, control hESCs (Day 8 shCtrl) efficiently produced neuroectodermal and mesendodermal lineages that contributed to the majority of clusters 3–6 (Fig. 2b, e, f). In stark contrast, the majority of ZBTB12 KD cells (Day 8 shZBTB12) failed to induce lineage differentiation and maintained a pluripotent state with a strong enrichment in cluster 1 (Fig. 2b, e, f). Trajectory analysis using two different algorithms, Monocle 3[40] and Slingshot[41], revealed two major branches on the UMAP plot toward neuroectodermal and mesendodermal lineages (Fig. 2g, Supplementary Fig. 3i). Pseudotime analysis on both trajectory algorithms confirmed that ZBTB12 KD strongly inhibited exit from pluripotency even after 8 days of FGF2 and TGFβ deprivation (Fig. 2h, Supplementary Fig. 3j). Furthermore, CytoTRACE, a computational framework for predicting differentiation states based on transcriptional diversity[42] suggested that cluster 1 dominated by Day 8 shZBTB12 cells was maintained as the least differentiated state along with clusters 0 and 2. (Fig. 2i, Supplementary Fig. 3k–m). Finally, we tested colony-initiating capacity after differentiation. Day 8 shCtrl cells failed to produce AP-positive colonies. Remarkably, Day 8 shZBTB12 cells were able to produce a significant number of AP-positive colonies (Supplementary Fig. 3n). Taken together, these results clearly show that ZBTB12 is essential for successful exit from a pluripotent state.

## ZBTB12-mediated suppression of stem cell dedifferentiation toward a naïve-like state

Conventional hPSCs are known to resemble primed PSCs in postimplantation embryos[1,2]. Although ZBTB12 KD cells maintained a pluripotent state after 8 days of differentiation, they constituted the majority of cluster 1 distinct from undifferentiated hESCs that are enriched in clusters 0 and 2 (Fig. 3a, b). To further characterize the cells in cluster 1, we conducted DEG analysis and identified 180 DEGs (100 genes with $\log_2 FC > 0.25$ and 80 genes with $\log_2 FC < -0.25$) (Fig. 3c, Supplementary Data 3). Compared to clusters 0 and 2, cluster 1 showed significant upregulation of naïve pluripotency-related genes such as NANOG, ESRG, UTF1, DPPA5, and endogenous retroviruses (ERVs) (Fig. 3c, d). Elevated expression of ERVs and ERV-related transcripts is a key feature of naïve PSCs[14,24]. Furthermore, significantly downregulated genes included WNT antagonists, SFRP1 and SFRP2, and primed pluripotency-related genes such as ZIC2 (Fig. 3c, d). Decreased expression of WNT antagonist is in agreement with the fact that WNT activation performs important roles in establishing a naïve pluripotent state[43]. GO terms of genes downregulated in cluster 1 (Supplementary Data 3) showed differentiation and developmental processes such as brain development, muscle structure development, cardiac chamber development, and respiratory system development (Supplementary Fig. 4a). This finding suggests that cells in cluster 1 are developmentally more primitive than those in clusters 0 and 2. Comparison of the expression of naïve and primed signatures from a recent study[44] further supported that cluster 1 exhibited a naïve-like gene expression profile (Fig. 3e). However, ZBTB12 KD cells did not show a

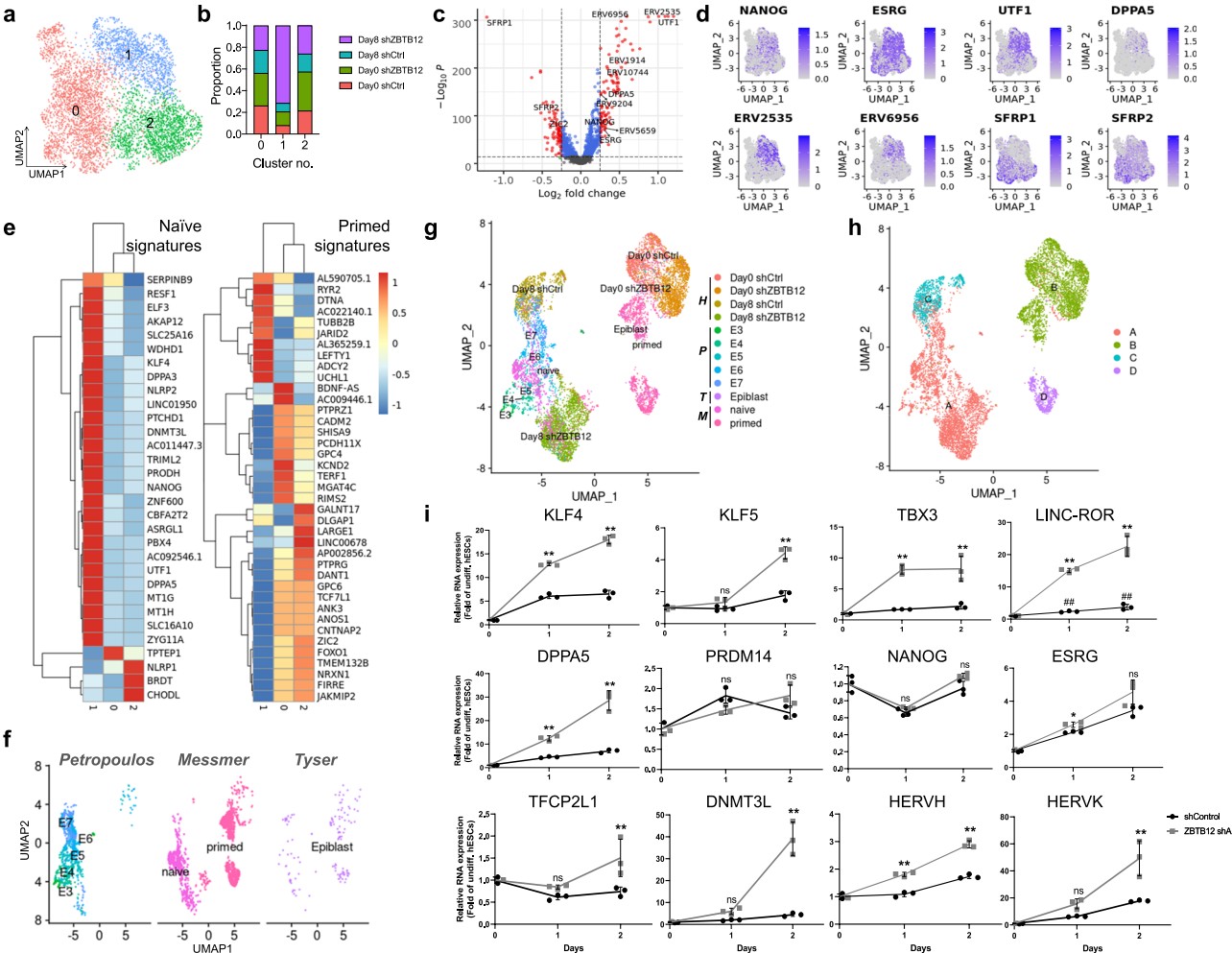

**Fig. 3 | ZBTB12 suppression drives hESCs toward a naïve-like state upon differentiation. a** Reconstructed UMAP of pluripotent stem cell clusters 0, 1, and 2 from Fig. 2. **b** Cellular compositions of each cluster. **c** Volcano plot showing differentially expressed genes (DEGs, red dots) in cluster 1 (compared to cluster 0 and 2). MAST algorithm is used to statistically determine DEGs, with Benjamini–Hochberg-corrected $p$ values < 0.05. Red dots indicate differentially expressed genes (or transcripts) with cutoff values for DEGs: $-\log_{10}P > 15$; $\log_2$ fold change < −0.25 or >0.25. Full list of DEGs can be found in Supplementary Data 3. **d** Expression of representative DEGs associated with naïve-state (NANOG, ESRG, UTF1, and DPPA5), ERVs (ERV2535 and ERV6956), and Wnt antagonists (SFRP1 and SFRP2) on UMAP. **e** Expression of naïve and primed signature genes[44] in each

cluster. **f** Split UMAP plots of the integrated three published datasets[45–47]. **g** UMAP plots of integrated scRNA-seq datasets (three from published datasets: P, Petropoulos, T, Tyser, M, Messmer and the scRNA-seq data of the current study: H, Han). **h** UMAP with new annotation with redefined clusters of Fig. 3g. **i** qPCR analysis of naïve markers in H9 hESCs expressing either control or ZBTB12 shRNA upon naïve induction with LIF and 3i (BIO, Dorsomorphin, PD0325901). Error bars represent mean ± sd ($n = 3$). Student's $t$ test (Holm–Sidak's multiple unpaired) *$p < 0.05$ and **$p < 0.01$ compared to shControl. ##$p < 0.01$ compared to Day 0 in LINC-ROR plot. ns not significant. All data points and the exact $p$-values can be found in the Source Data file.

formative transcriptional profile, assessed by marker gene expression as well as gene set enrichment score (Supplementary Fig. 4b–d).

To further validate the developmental status of ZBTB12 KD cells at day 8, we compared our scRNA-seq data (Clusters 0, 1, 2) in parallel with three other published datasets: Petropoulos et al. (2016, pre-implantation stage E3–E7)[45], Messmer et al. (2019, naïve and primed hESCs)[46] and Tyser et al. (2021, gastrulating human embryo)[47]. Most cells from pre-implantation human embryos (Petropoulos) were clustered together with naïve hESCs (Messmer). Epiblasts cells (Tyser) were located across naïve and primed clusters with a slight enrichment in primed clusters (Messmer), which indicates transitioning from a naïve to a primed pluripotent state (Fig. 3f, g). When all cells from three references and Fig. 3a were plotted together, Day 0 undifferentiated cells were clustered closely to the primed cells from Messmer et al. (Fig. 3g), suggesting these cells are mostly in a primed state as expected. In stark contrast, Day 8 shZBTB12 cells are closely located to the naïve (Messmer et al.) and E5 and E6 pre-implantation (Petropoulos et al.) cells, supporting the naïve-like feature of Day8 shZBTB12 cells

(Fig. 3g). Day 8 shCtrl cells were distinctly clustered from naïve or primed (Messmer et al.) and closely to E7 (Petropoulos et al.) cells (Fig. 3g). When we redefined clusters (cluster A to D) using the all-combined dataset, Day 8 shCtrl cells clustered distinctly (cluster C) from naïve-, pre-implantation, and Day 8 shZBTB12 clusters (cluster A) (Fig. 3h). Day 0 cells were clustered together with the major population of primed (Messmer et al.) cells (Fig. 3h). Furthermore, we applied weighted correlation network analysis (WGCNA)[48] to our pluripotent clusters to identify gene network modules in an unsupervised way. Cluster dendrogram identified three gene correlation network modules (brown, turquoise, and blue) (Supplementary Fig. 4e). Cluster 1 had the highest blue module eigengene score compared to cluster 0 and 2, indicating that the genes in the blue module either characterize or correlate with the cell state of cluster 1 (Supplementary Fig. 4f, g). Surprisingly, many ERV1 elements (including LINC-ROR), naïve marker genes (NANOG, DPPA5, and UTF1) and genes found in the naïve gene set (UTF1, DPPA5, MT1G, MT1H, and NANOG)[44] were found in the blue module, further supporting the idea of naïve-like transition of ZBTB12

KD cells after differentiation induction (Supplementary Fig. 4h, Supplementary Data 4). Increased expression of naïve marker genes was also observed in differentiating embryoid bodies after ZBTB12 KD (Supplementary Fig. 4i). Altogether, these results strongly indicate dedifferentiation of ZBTB12 KD cells from a primed to a naïve-like pluripotent state upon spontaneous differentiation induction.

Capturing in vitro naïve pluripotency was achieved in mouse ESCs by the inhibition of two kinases, mitogen-activated protein kinase and glycogen synthase kinase 3[49]. Treatment of LIF and two kinase inhibitors (2i) is a gold standard method to drive a naïve pluripotent state in mESCs and many culture conditions for human naïve pluripotency are also based on this formula. To test if ZBTB12 plays a role in the transition between naïve and primed pluripotent states, we transiently induced a naïve-like state in hESCs by a well-characterized culture medium condition including LIF and three inhibitors (3i, PD0325901, BIO, and Dorsomorphin)[50]. Treating hESCs with LIF and 3i was sufficient to transiently activate the expression of naïve pluripotency-related genes and ZBTB12 KD further promoted naïve gene activation (Fig. 3i). Consistent with a previous study[51], HERVK was more dramatically activated compared to mild induction of HERVH in a 3iL-induced naïve-like state, suggesting the possibility that HERVK and HERVH mark different naïve-like states induced either by chemicals or by ZBTB12 KD. However, ZBTB12 KD further enhanced the induction of both HERVK and HERVH (Fig. 3i). In contrast, ZBTB12 overexpression strongly suppressed the transcriptional transition towards a naïve-like state (Supplementary Fig. 4j). To test if the results were restricted to a specific naïve medium condition, we exploited a recently developed naïve-inducing medium condition that is based on CDK8/19 inhibition (CDK8/19i)[52] and confirmed the role of ZBTB12 in suppressing the primed to naïve transition (Supplementary Fig. 4k). Collectively, these results suggest that ZBTB12 serves as a molecular barrier for a naïve-like pluripotent state and this role of ZBTB12 may be essential for successful differentiation of hESCs. In the absence of ZBTB12, differentiating hESCs aberrantly enter a naïve-like state and thereby fail to activate lineage fates.

## ZBTB12-mediated fine-tuning of transcriptionally active HERVH loci

To explore the molecular mechanisms of how ZBTB12 regulates pluripotent state maintenance and differentiation, the genome-wide binding profile of ZBTB12 in hESCs was analyzed by chromatin immunoprecipitation with sequencing (ChIP-seq). Due to the lack of ChIP-seq grade antibodies, tagged ZBTB12 proteins were pulled down, resulting in 3621 ZBTB12 binding peaks (Supplementary Data 5). De novo motif prediction from our ChIP-seq data revealed that the ZBTB12 binding motif is highly similar to the previously reported motif identified in HEK293 cells[53], indicating the reliability of our ChIP-seq results (Supplementary Fig. 5a, Supplementary Data 6). In sharp contrast to the distribution profile of RNA Polymerase II binding sites enriched in promoter regions (Supplementary Fig. 5b), the majority of ZBTB12 binding sites were distributed in intergenic or intronic regions (Fig. 4a). These results suggest a potential role of ZBTB12 as a transcriptional regulator of non-coding regions. Indeed, ZBTB12 binding sites were significantly enriched for LTR retrotransposons, but not for other types of transposons and repeats (Fig. 4b). These results were confirmed when only ZBTB12 peaks containing the ZBTB12 motif were used (Supplementary Fig. 5c). LTR enrichment of ZBTB12 binding sites was also observed in HEK293 cells (Supplementary Fig. 5d)[53]. Furthermore, we found enriched motifs for OCT4, NANOG, and SOX2 within ZBTB12 binding sites (Supplementary Fig. 5e, Supplementary Data 6), suggesting that ZBTB12 is involved in the core pluripotency transcriptional network, specifically in LTR regulation.

Given that BTB-domain containing zinc finger proteins (BTB-ZFPs) generally function as transcriptional repressors via BTB/POZ domain-mediated protein interactions[54], we hypothesized that ZBTB12 would

function as a repressor for LTR expression. To test if ZBTB12 has global repressive effects on LTR expression in hESCs, bulk RNA-sequencing analysis was conducted after ZBTB12 KD. Among retrotransposons, ZBTB12 KD significantly elevated the expression of LTR ($D = 0.16$ and $p < 2.2e-16$ compared to total background, Kolmogorov–Smirnov (KS) test) and ERV1 subfamily members in particular ($D = 0.35$ and $p < 2.2e-16$ compared to total background, KS test) (Fig. 4c). Genome-wide transcriptional elevation of ERV1 by ZBTB12 KD was driven predominantly by the LTR7/HERVH subfamily, as the majority of LTR7/HERVH loci showed increased expression (D = 0.48 and $p = 2.2e-16$ compared to ERV1, KS test) (Fig. 4d, e, Supplementary Fig. 5f, g, h). These results were further confirmed by qPCR analysis after KD or overexpression of ZBTB12 in hPSCs with primers detecting LTR7 and HERVH GAG/POL sequences (Fig. 4f, g, Supplementary Fig. 5i, j, Supplementary Data 7). While it is not yet clear how ZBTB12 specifically regulates HERVH, ZBTB12 showed no significant genome-wide effect on the expression of protein coding genes (Supplementary Fig. 5k, l and Supplementary Data 8). Among full-length HERVH loci, those with high expression in hESCs (200 highest versus 200 lowest expression, $p = 1.153e-07$) exhibited low divergence time ($p = 2.2e-16$) (Supplementary Fig. 5m, n), implicating more active transcription of less-diverged HERVH loci[55,56]. Furthermore, HERVH loci with a large fold-change (top 200 loci) after ZBTB12 KD showed lower divergence time than the ones with a small fold-change (bottom 200 loci) ($p = 7.953e-10$) (Supplementary Fig. 5o). These results suggest that ZBTB12 is engaged in regulating less-diverged and transcriptionally active HERVH loci.

High expression of LTR7/HERVH is a key feature of hESCs, and this expression rapidly decreases as hESCs undergo transition from a naïve through a primed to a differentiated state (Supplementary Fig. 5p). Consistent with the repressive role of ZBTB12 in LTR7/HERVH, ZBTB12 KD strongly impeded downregulation of LTR7/HERVH expression during hESC differentiation (Fig. 4h). Furthermore, ZBTB12 depletion further enhanced LTR7/HERVH expression upon transient induction of naïve pluripotency by LIF and 3i (Supplementary Fig. 5q). Interestingly, the ZBTB12 protein levels resemble HERVH expression during stem cell differentiation with a naïve-like hESCs showing the highest ZBTB12 protein and HERVH RNA levels (Supplementary Fig. 5r). These results suggest that ZBTB12 contributes primarily to fine-tuning the expression of transcriptionally active HERVH loci and that downregulation of HERVH during stem cell differentiation is likely driven by other factors.

One important aspect regarding ZBTB12 control over LTR7/HERVH is that ZBTB12 is a highly conserved gene in vertebrates with a few amino acid changes between mouse and human (Ka/Ks = 0.0174919, p-Value (Fisher) = 3.63453e-90)[57] (Supplementary Fig. 6a, b), while its target, HERVH, invaded the human genome during primate evolution[23]. In accord with the sequence conservation, mouse ZBTB12 was able to suppress the expression of LTR7/HERVH that is absent in the mouse genome (Supplementary Fig. 6c). In contrast to hESCs, Zbtb12 KD in mouse epiblast stem cells (mEpiSC) did not alter global transcription of mouse retrotransposons including LINE, SINE, and LTR (Supplementary Fig. 6d, e, Supplementary Data 9), nor the differentiation capacity (Supplementary Fig. 6f–j). These results suggest that ZBTB12 acquired the new function during the primate evolution.

The core pluripotency factors, OCT4 and NANOG, have been implicated in activating LTR7/HERVH expression in hESCs[14,18,58]. Because enriched motifs for OCT4 and NANOG were observed in ZBTB12 binding sites (Supplementary Fig. 5e), we hypothesized that the opposed regulation between ZBTB12 and core pluripotency factors could contribute to fine-tuning the expression of LTR7/HERVH in hESCs. KD experiments showed that NANOG was a stronger activator of LTR7/HERVH than OCT4 (Supplementary Fig. 7a, b). NANOG overexpression significantly increased LTR7/HERVH expression (Supplementary Fig. 7c). Published ChIP-seq data[59] showed enriched NANOG peaks on LTR retrotransposons among transposons and repeat

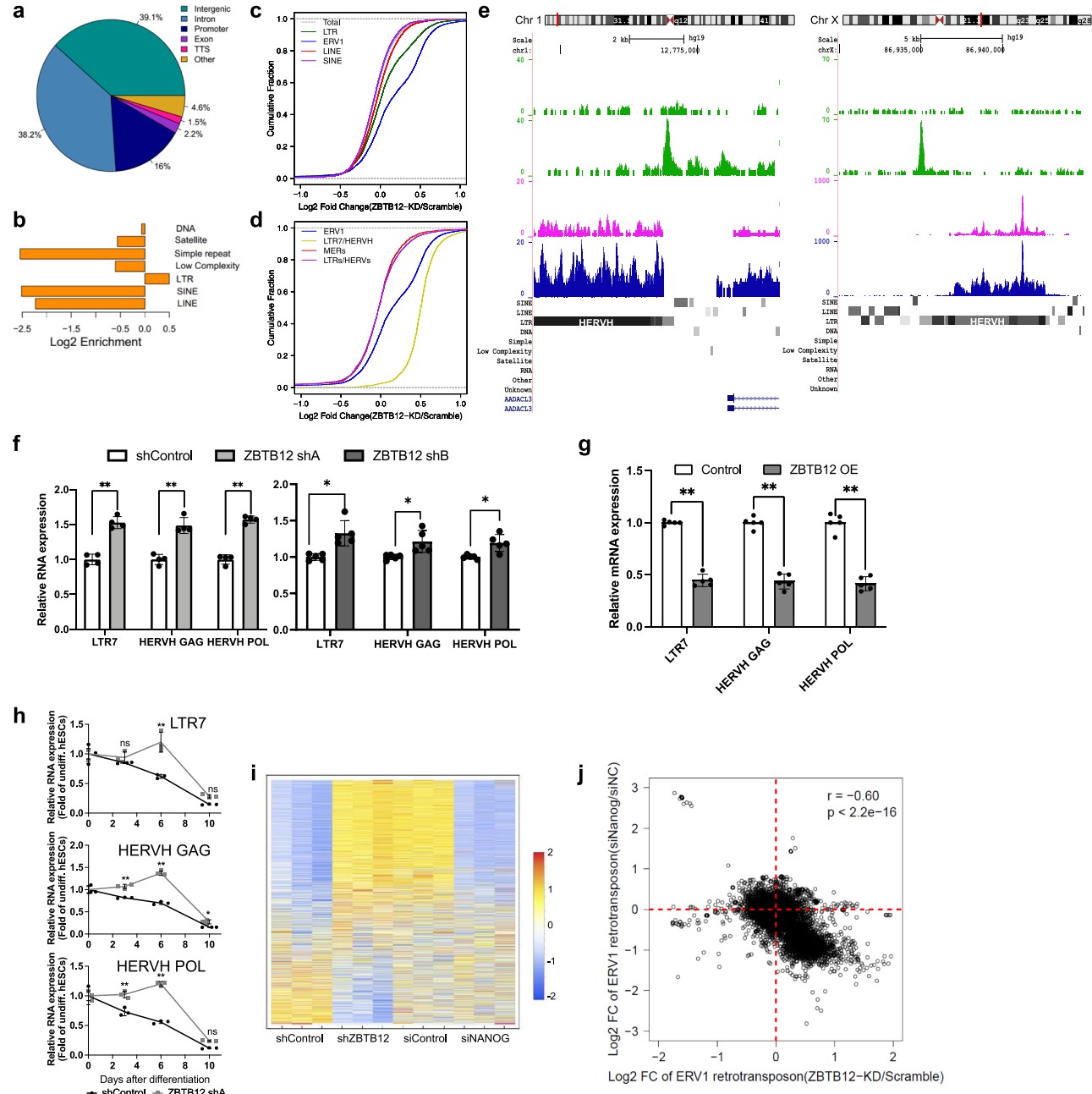

**Fig. 4 | ZBTB12 fine-tunes the expression of transcriptionally active HERVH loci. a** Genomic categories of ZBTB12 binding peaks. **b** Significant LTR enrichment of ZBTB12 ChIP-seq peaks among repetitive elements. **c** Log2 Fold Change of retrotransposon subtypes including LINE ($D = 0.070502$, $p$-value < 2.2e−16), SINE ($D = 0.014083$, $p$-value = 7.55e−15), LTR ($D = 0.16187$, $p$-value < 2.2e−16), and ERV1 ($D = 0.34723$, $p$-value < 2.2e−16) after ZBTB12 knockdown (Kolmogorov−Smirnov (KS) test for each subtype compared to total background, $n = 284397$ for Total; $n = 21267$ for LTR; $n = 82450$ for LINE; $n = 122210$ for SINE; $n = 8223$ for ERV1). **d** Log2 Fold Change of ERV1 ($D = 0.31112$, $p$-value < 2.2e−16) retrotransposon subtypes including LTR7/HERVH ($D = 0.79215$, $p$-value <2.2e−16), MERs and other LTRs/HERVs ($D = 0.028009$, $p$-value = 0.1319) after ZBTB12 knockdown (Kolmogorov−Smirnov (KS) test for each subtype compared to MERs subtype, $n = 8223$ for ERV1; $n = 3085$ for LTR7/HERVH; $n = 2529$ for MERs; $n = 5138$ for LTRs/

HERVs). LTRs/HERVs indicate ERV1 subtypes excluding LTR7/HERVH and MERs. **e** Examples of direct binding (ChIP-seq peak) and transcriptional regulation (bulk RNA-seq) of LTR7/HERVH loci by ZBTB12, displayed on UCSC genome track (http://genome.ucsc.edu). **f, g** qPCR analysis of LTR7, HERVH-GAG, HERVH-POL in undifferentiated H9 cells after ZBTB12 KD ($n = 4$ for ZBTB12 shA, $n = 5$ for ZBTB12 shB) (**f**) or OE ($n = 5$) (**g**). Error bars represent mean ± sd. Student's $t$ test (Holm−Sidak's multiple unpaired) *$p < 0.05$ and **$p< 0.01$. **h** qPCR analysis of LTR7, HERVH-GAG, HERVH-POL in H9 hESCs expressing either shControl or ZBTB12 shA after FGF2 and TGFβ deprivation. Data were normalized to Day 0. Error bars represent mean ± sd ($n = 3$). Student's $t$ test (Holm−Sidak's multiple unpaired) *$p < 0.05$ and **$p < 0.01$ compared to shControl. ns non-significant. **i, j** Heatmap (**i**) and scatter plot (**j**) of ERV1 expression after either ZBTB12 KD or NANOG KD ($n = 3$ each). All data points and exact $p$-values can be found in the Source Data file.

sequences (Supplementary Fig. 7d). By bulk RNA-seq analysis, we confirmed that NANOG KD induced global downregulation of ERV1 family members ($D = 0.36$ and $p = 2.2$e−16 compared to total background, KS test) and LTR7/HERVH in particular ($D = 0.53$ and $p = 2.2$e

−16 compared to ERV1, KS test) (Supplementary Fig. 7e, f). Furthermore, ERV1 family members showed a significant inverse correlation in expression fold-change between NANOG KD and ZBTB12 KD (Pearson correlation analysis, $R = −0.60$, $p < 2.2$e−16) (Fig. 4i, j, Supplementary

Data 10). Although the detailed mechanisms of NANOG and ZBTB12 interaction are unclear, these results suggest that both ZBTB12 and NANOG contribute to fine-tuning the expression level of LTR7/HERVH in hESCs.

## Molecular features of ZBTB12-mediated suppression on primate specific LTR7/HERVH

Given the evolutionary conservation of ZBTB12, it is unlikely that ZBTB12 directly recognizes the sequence of newly invaded HERVHs. Indeed, among 70 ZBTB12 ChIP-seq peaks associated with full length HERVH loci (Supplementary Data 11), only three peaks were located on LTR7. The majority of ZBTB12 peaks were found within 10-kb up- or downstream of LTR7 (Supplementary Fig. 8a). These results suggest that the global transcriptional activation of HERVH loci can be attributed to both direct and indirect effects of ZBTB12 KD in hESCs. KAP1 is a key epigenetic corepressor that mediates silencing of ERVs by direct interaction of KRAB-ZFPs. However, we found that ZBTB12 did not bind to KAP1 (Supplementary Fig. 8b, c). ZBTB16 (also known as PLZF) is one of the most studied members of the BTB-ZFP family and a previous report showed a direct interaction between ZBTB16 and the SIN3A/HDAC co-repressor complex[60]. Therefore, we first analyzed the global co-occupancy of ZBTB12 with the SIN3A/HDAC complex in hESCs. Indeed, analysis of published ChIP-seq data[61–63] revealed that the SIN3A/HDAC co-repressor complex is highly enriched in ZBTB12 ChIP-seq peaks (Supplementary Fig. 8d). In addition, a large fraction of these loci overlapped with NANOG and H3K27ac, but not with H3K27me3, suggesting the cooperative function of ZBTB12, NANOG, and the SIN3A/HDAC complex in regulating transcriptionally active loci (Supplementary Fig. 8d). Co-immunoprecipitation assay confirmed that ZBTB12 proteins interacted with the SIN3A/HDAC complex (Supplementary Fig. 8e–g). We further conducted ChIP-seq using anti-H3K27ac and anti-HDAC1 antibodies after ZBTB12 KD in hESCs (Supplementary Fig. 9a-d, Supplementary Data 12). Similar to the previously published datasets, HDAC1 and H3K27me3 peaks were frequently found nearby ZBTB12 binding peaks. The entire ZBTB12 binding peaks were grouped into eight clusters (C1–C8) by k-means clustering (Supplementary Fig. 9a, b, Supplementary Data 12), revealing that 86.6% of ZBTB12 peaks associated with full-length LTR7/HERVHs are distributed in C1, C2, C5, and C6 where H3K27ac ChIP-seq reads are enriched (Supplementary Fig. 9a–c). Furthermore, 68.7% of ZBTB12 peaks associated with full-length LTR7/HERVHs are present in C1, C4, C6, and C7, where HDAC1 binding is enriched (Supplementary Fig. 9a–c). In line with our hypothesis, H3K27ac levels nearby ZBTB12 peaks were significantly increased after ZBTB12 KD (Supplementary Fig. 9d). However, HDAC1 showed no significant difference between control and ZBTB12 KD, which could be in part due to complex feedback regulation between epigenetic factors and HERVH (Supplementary Fig. 9d) (see discussion). We further tested if pharmacological inhibition of HDAC1 can revert the effect of ZBTB12 on LTR7/HERVH. Treatment of valproic acid, a HDAC inhibitor, was sufficient to prevent the ZBTB12-mediated suppression of LTR7/HERVH (Supplementary Fig. 9e). Although the precise molecular mechanisms are not clear, our results suggest that epigenetic mechanisms are involved in ZBTB12-mediated transcriptional suppression.

## Pluripotency exit by ZBTB12-mediated suppression of HERVH-overlapping lncRNAs

We next sought to investigate a molecular mechanism that bridges ZBTB12 control over LTR7/HERVH to exit from pluripotency. Recently, it was reported that transposable elements are major contributors for evolutionary emergence, diversification, and regulation of lncRNAs[32].

Focusing on LTR7/HERVH, we identified 280 lncRNAs overlapped with LTR7/HERVH loci in the human genome (Supplementary Data 13). Among them, 56 lncRNAs were detected in our RNA-seq data with counts per million (CPM) higher than one (Supplementary Data 14), which is similar to the results from a previous study[14]. Significant correlation of HERVH expression with the overlapping lncRNAs was observed after ZBTB12 KD (Pearson correlation analysis, $R = 0.53$, $p < 0.001$) (Fig. 5a). Increased expression of selected lncRNAs (LINC-ROR, ESRG, VLDLR-AS1, LMCD1-AS1, and LINC01356) by ZBTB12 KD was further validated by qPCR analysis (Fig. 5b, g, Supplementary Fig. 10a).

Among the 56 LTR7/HERVH-associated lncRNAs, LINC-ROR is the most induced lncRNA after ZBTB12 KD (Fig. 5a). Elevated LINC-ROR expression is also observed upon transient induction to a naïve-like state (Fig. 3i). Furthermore, ZBTB12 directly binds upstream of the LINC-ROR locus (Supplementary Fig. 10b). Exogenous expression of ZBTB12 decreased the expression of LINC-ROR (Supplementary Fig. 10c) and reverted ZBTB12 KD-driven increase (Supplementary Fig. 10d). Upon spontaneous differentiation by FGF2 and TGFβ deprivation, LINC-ROR expression gradually decreased (Fig. 5c). However, ZBTB12 KD retained LINC-ROR expression similar to that of undifferentiated hESCs even after 6 days of differentiation (Fig. 5c). Because ZBTB12 KD delayed pluripotency exit and retained prolonged expression of core pluripotency genes after differentiation (Fig. 1e, f), we hypothesized that ZBTB12-mediated suppression of LINC-ROR could be essential for successful pluripotency exit. Indeed, LINC-ROR KD was sufficient to reduce colony-initiating capacity and to facilitate pluripotency exit (Fig. 5d, e, Supplementary Fig. 10e). Furthermore, LINC-ROR overexpression phenocopied ZBTB12 KD with increased colony-initiating capacity (Supplementary Fig. 10f, g) and impaired exit from a pluripotent state (Fig. 5f, Supplementary Fig. 10h). LINC-ROR overexpression induced mild elevation of some naïve marker genes during differentiation (Supplementary Fig. 10i), suggesting that LINC-ROR is in part involved in hPSC dedifferentiation by ZBTB12 KD. More importantly, LINC-ROR suppression allowed rescue of the ZBTB12 KD phenotype by efficiently shutting down the core pluripotency genes during differentiation (Fig. 5j, Supplementary Fig. 10o). These results suggest that ZBTB12-mediated suppression of LINC-ROR is essential for successful exit from a pluripotent state.

ESRG is a HERVH-associated transcript that is highly expressed in hESCs[14]. The elevated level of ESRG is also used as a marker for naïve-like hESCs[14]. However, the exact role and mechanisms of ESRG in pluripotency are not clear. ZBTB12 peaks were found in ESRG locus in a subthreshold level suggesting that ESRG could be an indirect target of ZBTB12 (Supplementary Fig. 10j). ZBTB12 KD induced mild elevation of ESRG expression in hPSCs (Fig. 5a, g, Supplementary Fig. 10j). Despite the high expression level in hESCs, ESRG KD did not have any significant effects on the expression of core pluripotency genes (Supplementary Fig. 10k, l). These results are in agreement with a recent report showing that ESRG is dispensable for hPSC self-renewal[64]. Upon differentiation, however, ESRG KD boosted dissolution of the core pluripotency gene network (Fig. 5h, Supplementary Fig. 10m), thereby facilitating lineage specification towards neuroectodermal and mesendodermal fates (Fig. 5i, Supplementary Fig. 10n). Similar to LINC-ROR, ESRG KD rescued the ZBTB12 depletion phenotype in pluripotency exit (Fig. 5j, Supplementary Fig. 10o). Altogether, these data suggest that ZBTB12 safeguards lineage differentiation by efficiently shutting down the LTR7/HERVH-overlapping lncRNAs.

## Discussion

The pluripotent state transition from a naïve through a formative to a primed state accompanies dynamic changes in self-renewal ability and differentiation potential. Naïve PSCs are characterized by elevated self-renewal ability and blunted differentiation potential, while the formative transition enables lineage capacitation. Therefore, it has been

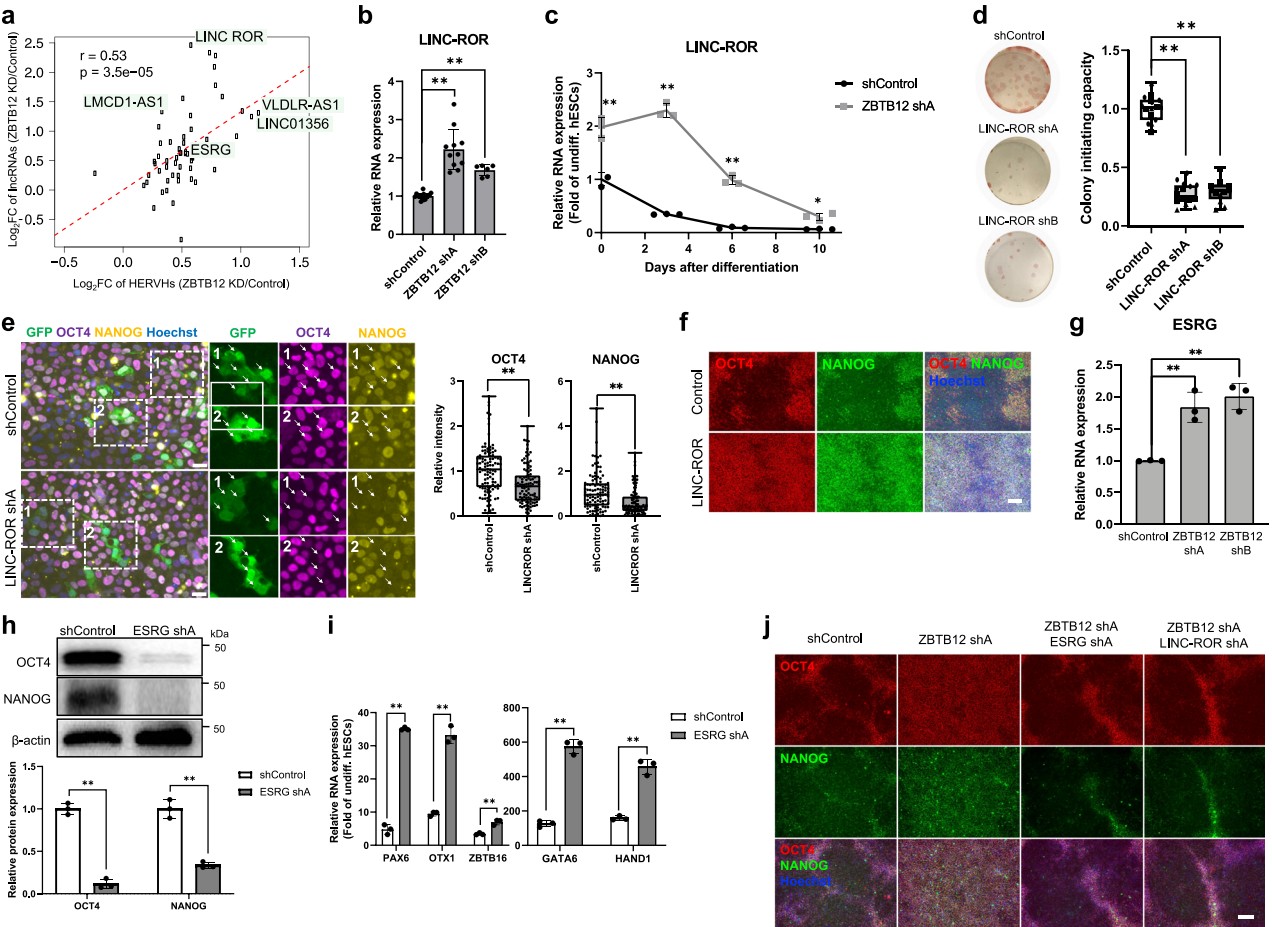

**Fig. 5 | Suppression of HERVH-overlapping lncRNAs by ZBTB12 drives pluripotency exit. a** Correlation of HERVH expression with the overlapping lncRNAs after ZBTB12 KD in H9 hESCs. Pearson's correlation test. **b** qPCR analysis of LINC-ROR in H9 hESCs after ZBTB12 KD. $n = 17$ for shControl, $n = 11$ for ZBTB12 shA, and $n = 6$ for ZBTB12 shB. Student's $t$ test (two-tailed unpaired) **$p < 0.01$ compared to shControl. **c** qPCR analysis of LINC-ROR in shControl- or ZBTB12 shA-transduced H9 hESCs during differentiation by FGF2 and TGFβ deprivation ($n = 3$). Student's $t$ test (Holm−Sidak's multiple unpaired). *$p < 0.05$ and **$p < 0.01$ compared to shControl. **d** Colony forming assay in undifferentiated H9 hESCs transduced with lentiviral vectors expressing LINC-ROR shRNAs. Representative AP-stained wells are shown. $n = 18$ for shControl and shA from three independent experiments and $n = 12$ for shB from two independent experiments. Independent experiments are represented by different point shapes. Student's $t$ test (two-tailed unpaired). **$p < 0.01$ compared to shControl. **e** Immunofluorescence assay for OCT4 and NANOG in differentiated (day 5) H9 hESCs transduced with lentiviral vectors expressing EGFP-shControl or EGFP-LINC-ROR shA. Arrows indicate GFP positive cells. $n = 100$ from three independent experiments. Scale bar, 25 μm. Student's $t$ test (two-tailed unpaired) **$p < 0.01$. Data are represented as box-plot with

median and min to max with all data point. Center line, median; box limits, upper and lower quartiles; whiskers, max and min; points, all data points (**d**, **e**). **f** Representative images from three independent immunofluorescence assays for OCT4 and NANOG in differentiated (day 5) H9 hESCs transduced with lentiviral vectors expressing LINC-ROR. Scale bar, 100 μm. **g** qPCR analysis of ESRG expression in H9 hESCs expressing either control or ZBTB12 shRNAs ($n = 3$). Student's $t$ test (two-tailed unpaired) **$p < 0.01$. **h** Western blots of OCT4 and NANOG in differentiated (day 5) H9 hESCs transduced with lentiviral vectors expressing either shControl or ESRG shA ($n = 3$). Student's $t$ test (Holm−Sidak's multiple unpaired) **$p < 0.01$. **i** qPCR analysis of neuroectoderm (PAX6, OTX1, ZBTB16) and mesendoderm (GATA6, HAND1) markers in differentiated (day 8) H9 hESCs transduced with lentiviral vectors expressing shControl or ESRG shA ($n = 3$). Student's $t$ test (Holm−Sidak's multiple unpaired) **$p < 0.01$. **j** Representative images from three independent immunofluorescence assays for OCT4 and NANOG in differentiated (day 5) H9 hESCs transduced with lentiviral vectors expressing ZBTB12 shA in combination with ESRG shA or LINC-ROR shA. Scale bar, 100 μm. All data points and exact $p$-values can be found in the Source Data file.

suggested that naïve pluripotency exit in peri-implantation development is a process in which PSCs gradually acquire mature differentiation potential[1,2]. However, the underlying mechanisms for the acquisition of differentiation potential has not been fully understood. In this study, we found that ZBTB12 is a key molecular balancer between self-renewal and differentiation. ZBTB12 depletion boosted self-renewal of hPSCs at the expense of differentiation potential (Fig. 1c, e-h). Exogenous expression of ZBTB12 facilitated the exit from a pluripotent state (Supplementary Fig. 2q). Single cell transcriptomic analysis revealed that ZBTB12-depleted hPSCs unexpectedly acquired a naïve-like state upon differentiation, and this dedifferentiation event impaired pluripotency exit and three germ layer differentiation (Fig. 3). These results suggest that human pluripotent states are highly

metastable and that primed hPSCs require a molecular barrier to prevent dedifferentiation and to transit into developmentally advanced states. Interestingly, ZBTB12 is a conserved gene in vertebrates (Supplementary Fig. 6a) that evolutionarily gained a regulatory function in primate-specific LTR7/HERV (Fig. 4d). Efficient exit of hPSCs toward three germ layers is achieved by ZBTB12-mediated suppression of HERVH-overlapping lncRNAs (Fig. 5). Overall, our findings revealed the regulatory relationship between ZBTB12 and HERVH-overlapping lncRNAs as a species-specific mechanism for stem cell differentiation and dedifferentiation.

In developing mammalian embryos, PSCs exist transiently from a blastocyst to a gastrula and undergo continuous transition in a continuum of pluripotent states[1,2,11]. However, specific pluripotent states

can be stably captured in vitro by well-defined culture conditions[1,2]. In conventional hPSC culture, FGF2 and TGFβ/Activin play key roles in maintaining a primed pluripotent state[65,66]. Strong activation of these signaling pathways is known to suppress a naïve pluripotent state and block stem cell differentiation[49,65], thereby trapping cells in a primed state. Therefore, it is conceivable that upon withdrawal of FGF2 and TGFβ, hPSCs exit from a primed state and can proceed either backward to a naïve state or forward to a differentiated state. In the absence of ZBTB12, indeed, primed hPSCs failed to advance to differentiated states but were able to predominantly dedifferentiate toward naïve-like cells (Fig. 3). These findings suggest that differentiation is not a default pathway upon exit from a primed pluripotent state and that a molecular barrier for dedifferentiation is essential for successful differentiation.

Although early developmental TFs are evolutionarily well conserved, spatiotemporal regulation of the TFs varies significantly between species[67]. By creating new TF binding sites and lncRNAs, HERV invasion into the human genome has been proposed to make a significant contribution to rewiring transcriptional networks and to directing species-specific developmental programs[18–21]. In line with this idea, different HERV families are activated in a stage-specific manner during the earliest human embryonic development[22]. In particular, LTR7/HERVH, a primate-specific HERV family, possesses binding sites for key pluripotency TFs and shows high expression in blastocyst stages[18,22]. Likewise, in vitro naïve-like hPSCs exhibit high LTR7/HERVH expression, which gradually declines as hPSCs transit into a primed state (Supplementary Fig. 5p)[14]. Exit from a primed pluripotent state induces further downregulation of LTR7/HERVH expression (Supplementary Fig. 5p)[58]. Consistently, we found that upregulation of HERVH-overlapping lncRNAs by ZBTB12 KD delayed pluripotency exit from a primed pluripotent state and thus impeded lineage derivation. Furthermore, depletion of HERVH-overlapping lncRNAs was sufficient to promote the exit from a pluripotent state (Fig. 5e, h). These data suggest that LTR7/HERVH serves as a key molecular regulator of human stem cell states. Precise control of dynamic stem cell states demands regulatory mechanisms for fine-tuning of LTR7/HERVH expression. Among core pluripotency genes, NANOG showed a substantial effect on promoting LTR7/HERVH expression (Supplementary Fig. 7). By analyzing expression fold-changes of individual LTR7/HERVH loci upon ZBTB12 KD and NANOG KD, we observed significant inverse correlation, indicating opposed effects of ZBTB12 and NANOG on the expression of LTR7/HERVH (Fig. 4i, j, Supplementary Data 10). Together with previous reports, our findings propose that LTR7/HERVH is a species-specific molecular rheostat of dynamic human pluripotent states.

We tested two HERVH-overlapping lncRNAs based on the highest log2FC after ZBTB12 KD (LINC-ROR) or the highest basal expression level (ESRG) as downstream effector candidates. LINC-ROR was reported to post-transcriptionally regulate the expression of core pluripotency genes, OCT4, NANOG, and SOX2, by acting as a miRNA sponge[68]. Accordingly, LINC-ROR KD was sufficient to reduce colony-initiating capacity and to facilitate pluripotency exit, while LINC-ROR overexpression prevented pluripotency exit (Fig. 5d–f). On the other hand, there is a discrepancy between studies on the role of ESRG in pluripotency[14,64] and the mechanism is underexplored. In the current study, we found that ESRG KD had no effect on the expression of core pluripotency genes in hESCs (Supplementary Fig. 10k, l) in line with Takahashi et al.[64]. However, upon spontaneous differentiation, ESRG KD facilitated the pluripotency exit and differentiation (Fig. 5h, i), supporting the results from Wang et al.[14]. Detailed modes of action of ESRG need to be further studied. Although KD of LINC-ROR or ESRG partially rescued the ZBTB12 KD-driven pluripotency exit delay (Fig. 5j), we are doubtful that these two lncRNAs are solely attributable to the function of ZBTB12. Given that almost all lncRNAs associated to HERVH showed elevated expression after ZBTB12 KD, it is plausible

that multiple HERVH-overlapping lncRNAs orchestrate the regulation of pluripotency.

An evolutionary arms race between the KRAB-ZFP family and ERVs is a well-documented mechanism through which host cells have been taming genetic invaders. A rapid expansion of KRAB-ZFP family genes in primates resulted in more than 350 members with evolutionarily selected mutations at zinc finger motifs. In a continuous arms race, the emergence of a new KRAB-ZFP gene suppresses target ERVs, and this suppression drives the evolution of new ERVs escaping KRAB-ZFP binding[17]. However, this mechanism raises a critical question: how did host cells deal with the immediate dangers induced by new ERV invasion before the evolution of new KRAB-ZFP genes? To our knowledge, this is the first report to demonstrate the role of a BTB-ZFP in ERV regulation. KRAB-ZFPs are restricted to tetrapod vertebrates and have undergone dramatic expansion and diversification in mammals[69]. In contrast, BTB-ZFPs are found in all animal species and ZBTB12 is well-conserved in vertebrates[69]. Therefore, ZBTB12-mediated suppression of primate-specific HERVH suggests a distinct regulatory mechanism that is outside of an evolutionary arms race. Furthermore, ZBTB12 shows two major differences in ERV regulation compared to KRAB-ZFPs. First, ZBTB12 proteins bind the SIN3A/HDAC complex (Supplementary Fig. 8e–g), while KAP1 corepressor is predominantly engaged in KRAB-ZFP-mediated transcriptional suppression. Second, contrary to some KRAB-ZFPs that are reported to directly recognize primer binding sites (PBSs) of ERVs[17,70], ZBTB12 binding sites were frequently found in up- or downstream of HERVH loci (Supplementary Fig. 8a). This finding suggests two possibilities: (1) new ZBTB12 binding sites were evolved nearby HERVH loci to control its expression and/or (2) HERVH insertions nearby pre-existing ZBTB12 binding sites were positively selected during evolution. Taken together, our study unveiled a new regulatory mechanism of ERV domestication and its role in stem cell differentiation.

Although ZBTB12 was shown to interact with HDAC1 and SIN3A and ZBTB12 KD increased the H3K27ac level in ZBTB12 binding sites, HDAC1 binding nearby ZBTB12 binding loci was not altered by ZBTB12 KD (Supplementary Fig. 9d). This result may be explained by previous findings on epigenetic roles of LTR7/HERVHs. First, transcriptionally active HERVH loci are reported to demarcate topologically associating domains (TADs) in hPSCs[71], where dynamic epigenetic modifications including HDAC1-driven deacetylation of H3K27ac occur[72]. Therefore, the elevated HERVH transcription by ZBTB12 KD may result in changes in TADs and in the binding of epigenetic modulators. Secondly, the SIN3A/HDAC corepressor complex has been shown to directly bind with NANOG to promote pluripotency[73]. Given that our ZBTB12 ChIP-seq loci are highly enriched with the NANOG-binding motif (Supplementary Fig. 5e), it is likely that HDAC1 recruitment to the ZBTB12 binding loci is dynamically regulated by multimodal mechanisms. Therefore, detailed molecular mechanisms for ZBTB12-meidated HERVH suppression needs further investigation.

Cellular dedifferentiation has been increasingly observed in several pathological conditions including cancer and infectious diseases[28,74] as well as embryonic development such as neural crest cell specification[15]. However, precise molecular mechanisms driving cellular dedifferentiation remain unknown. Discovery of ZBTB12 as a molecular barrier for dedifferentiation suggests that inactivation of molecular barriers could be an important mechanism for cellular dedifferentiation. This conceptual idea will provide a new perspective in understanding human diseases where cellular identity is lost.

## Methods

### Cell lines

H1 (male), H9 (female) hESCs, and IMR90 (female) hiPSCs were purchased from WiCell and maintained in feeder-free conditions on Matrigel (Corning) either in mTeSR1 (Stem Cell Technologies) or TeSR-E8 (Stem Cell Technologies) medium. Cells were authenticated by

short tandem repeat analysis. Cells were passaged every 5–6 days by ReLeSR (Stem Cell Technologies), or by Accutase (ThermoFisher Scientific). This work was approved by the Human Stem Cell Research Oversight Committee at Pohang University of Science and Technology (PIRB-2021-R035). HEK293T was maintained in DMEM supplemented with 10% fetal bovine serum (FBS). Every 3-4 days cells were trypsinized and plated onto new dish to avoid high confluency.

J1 mouse epiblast stem cells (mEpiSCs) were maintained on Matrigel (Corning) in a primed culture medium (DMEM/F-12 supplemented with 20% knockout serum replacement, MEM nonessential amino acid solution, GlutaMAX and bFGF (Peprotech, 10 ng/ml), Activin A (Peprotech, 20 ng/ml)). Cells were passaged every 4–5 days by ReLeSR (Stem cell Technologies).

### Lentiviral preparation and concentration
Lentiviral vector plasmids (Supplementary Table 2) were transfected into HEK293T cells with packaging plasmids psPAX2 (Addgene #12260) and pMD2.G (Addgene #12259). Supernatants were collected 48 h post-transfection and filtered through a 0.45-µm filter. Viral supernatants were concentrated using Lenti-X Concentrator (Takara), resuspended in PBS and stored at −80 °C.

### Inducible ZBTB12 KD in hESCs
H9 hESCs were transduced by a lentiviral vector expressing IPTG-inducible shControl or shZBTB12 and then selected with puromycin (1 µg/ml) for 48 h post transduction. Puromycin-selected cells were treated with IPTG (ThermoFisher scientific) (1 mM) for 3 days and then were differentiated in a spontaneous differentiation medium (DMEM/F12, 10–15% knockout serum replacement, MEM nonessential amino acid solution and 0.1 mM β-mercaptoethanol) in the presence of IPTG (1 mM). To recover ZBTB12 expression, IPTG was withdrawn for 48 h before differentiation and then cells were induced to spontaneously differentiate without IPTG. Media were refreshed every other day.

### siRNA transfection
hESCs were transfected with siRNAs (negative control-12935300, NANOG-HSS188276 and OCT3/4-HSS143401, ThermoFisher Scientific) (100 nM) with RNAiMAX (ThermoFisher Scientific) according to the manufacturer's protocol.

### Colony formation assay
hESCs transduced with lentiviruses were dissociated into single cells with Accutase. 5000–10,000 cells were plated on Matrigel-coated 12-well plates with mTeSR1 or TeSR-E8 supplemented with Y-27632 (10 µM). For differentiated hESCs, 20,000 cells were plated with TeSR-E8 in the presence of Y-27632 (10 µM). From day 1, media without Y-27632 were refreshed every other day. Cells were fixed at day 8 and the number of colonies were measured after alkaline phosphatase staining using VECTOR® Red Alkaline Phosphatase (Red AP) Substrate Kit (VectorLabs, SK-5100).

### Spontaneous differentiation of hESCs
hESCs were induced to spontaneous differentiate on Matrigel in hESC culture medium (DMEM/F12, 10–15% knockout serum replacement, MEM nonessential amino acid solution and 0.1 mM β-mercaptoethanol) without FGF2. For the experiments with lentiviral transduction, cells were transduced at day −3, selected with puromycin (1 µg/ml) at day −1 for 24 h and harvested (day 0 samples) or induced for differentiation at day 0. Media were refreshed every other day.

### Neuroectoderm differentiation of hESCs
hESCs were induced to differentiate into neuroectoderm by dual SMAD inhibition (DMEM/F12, 15% knockout serum replacement, MEM nonessential amino acid solution and 0.1 mM β-mercaptoethanol with

SB 431542 (Peprotech, 10 µM) and Dorsomorphin dihydrochloride (Tocris, 2 µM). Media were refreshed every other day.

### Differentiation of mEpiSCs
J1 mEpiSCs were transduced with lentivirus expressing shControl or shZbtb12 and then selected with puromycin (2 µg/ml) for 48 h post transduction. When selected cells reached 70–80%, cells were dissociated with ReLeSR and cultured on ultra-low attachment plates (Corning) in a differentiation medium (DMEM/F-12 supplemented with 20% knockout serum replacement, MEM nonessential amino acid solution, GlutaMAX). Media were refreshed every other day. For neuroectoderm differentiation, retinoic acid (Sigma, 10 µM) was added in a differentiation medium (DMEM supplemented with 15% FBS, MEM nonessential amino acid solution, 0.1 mM β-mercaptoethanol and Glutamax). Media were refreshed every other day.

### Embryoid body formation
hESCs were transduced with Ctrl- or ZBTB12-shRNA lentiviruses and selected using puromycin (1 µg/ml) at 24 h post transduction. Once puromycin-selected cells reach 60–70% confluency, cells were dissociated using ReLeSR (Life Technologies) and suspension-cultured on ultra-low attachment plates (Corning Costar, Corning, NY) in spontaneous differentiation medium supplemented with Y-27632 (10 µM). Media were refreshed every other day without Y-27632.

### Induction of human naïve-like pluripotent states
H9 hESCs cultured on Matrigel-coated plates in mTeSR1 medium were induced to a naïve-like state by transferring to mTeSR1 medium containing LIF (10 ng/ml), PD0325901 (1 µM), BIO (1 µM), and Dorsomorphin (1 µM)[50]. For CDK8/19i-based induction of a human naïve-like state, H9 hESCs on Matrigel were cultured in mTeSR1 medium supplemented with LIF (20 ng/ml) and Senexin A (10 µM)[52].

### Immunofluorescence staining
Cells were fixed with 4% paraformaldehyde and permeabilized with 0.25% Triton X-100, followed by blocking with 10% FBS in PBS for 1 h. Samples were stained with primary antibodies for overnight at 4 °C. Secondary antibody staining was performed for 1 h at room temperature with Alexa Fluor 488-donkey anti-rabbit IgG, Alexa Fluor 555-donkey anti-goat IgG, Alexa Fluor 647-donkey anti-mouse IgG, Alexa Fluor 488-donkey anti-goat IgG and Alexa Fluor 555-donkey anti-mouse IgG (ThermoFisher Scientific). Hoechst staining was performed for 2 minutes at room temperature with Hoechst solution (ThermoFisher scientific, 62249). Images were taken using Leica SP8 con-focal microscope or Olympus IX71 fluorescence microscope.

### Quantitative real-time PCR
Total RNAs were extracted using TRIzol Reagent (ThermoFisher Scientific), followed by reverse transcription with SuperScript IV First-Strand Synthesis System (ThermoFisher Scientific). qPCR was performed with Power SYBR Green PCR Master Mix (Applied Biosystems) using QuantStudio 12 K Flex Real-Time PCR System. GAPDH was used as a normalization control. Primers used in this study are listed in Supplementary Table 3.

### Western blot
The same amount of protein samples was subjected to SDS-PAGE, followed by transfer to nitrocellulose membranes (GE Healthcare Life Sciences). PBST (0.1% Tween 20 in PBS) containing 5% BSA was used to block the membranes. Immunoblotting was performed overnight at 4 °C with primary antibodies (listed in Supplementary Table 4). The membranes were stained with secondary antibodies Alexa Flour 680-goat anti-mouse IgG (ThermoFisher Scientific) and IRDye 800CW-goat anti-rabbit IgG (LI-COR), or donkey anti-goat IgG (H + L)-HRP and goat anti-mouse IgG (H + L)-HRP (ThermoFisher Scientific) for 1 h at room

temperature. Protein bands were visualized with LI-COR Odyssey Imaging System (LI-COR). Uncropped original scan images are provided in Source Data file.

## Co-immunoprecipitation

Cells were lysed with IP-lysis buffer (50 mM Tris-HCl pH7.5, 150 mM NaCl, 0.5% TX-100 and 1 mM EDTA) in the presence of protease inhibitor cocktail (Roche) at 4 °C on rotator for 50 min. After centrifugation at 18,000×g at 4 °C for 15 min, supernatants were harvested and used immediately or stored at −80 °C. Protein concentration was measured using Pierce BCA Protein Assay Kit (Thermo-Fisher Scientific). Small portion (~40 µg) of protein lysates were stored for whole cell lysate control. Same amounts of lysates were then subjected to immunoprecipitation using Pierce™ Anti-HA Magnetic Beads (ThermoFisher, 88836) according to the manufacturer's protocol. Whole cell lysates and magnetic beads after immunoprecipitation were diluted in 2× Laemmli Sample Buffer (Biorad, 1610737) supplemented with 2-mercaptoethanol (Biorad, 1610710) and incubated at 95 °C for 5 min.

## NanoCAGE

To build nanoCAGE libraries, directed differentiation of H9 hESCs to cerebral cortex neurons was carried out as described[33]. Differentiating cells were harvested at day 0 (before differentiation), 5, 15, and 23 and total RNA is extracted for library preparation. Totally, 100 ng of RNA was used to prepare the nanoCAGE libraries with modified protocol based on published protocol[34]. To pool four samples for sequencing, we modified the template switching oligonucleotides with six nucleotides barcode sequence right before the three guanosine ribonucleotides (rG). As we used the NextSeq sequencing platform, we changed the 5′ and 3′ adapter sequences and sequencing primers which could be compatible with our sequencing platform. Different barcodes were used for the two replicates of each sample. Finally, eight libraries were sequenced using Illumina NextSeq 500 Mid Output or High Output Kit and single end 150 bp cycles, two replicates for each sample.

Reads of nanoCAGE data were assigned to each sample according to the barcode sequence with <= 2 mismatches. Then, the first 11 nucleotides of each read were trimmed. After trimming the reads, bwa software (version 0.7.13, mem default parameter) was used to align the read to human reference genome (hg19 version) from UCSC genome database. Comprehensive human gene annotation file (version 19) was downloaded from GENCODE database[75].

Gene expression was measured by HTSeq[76] and normalized by CPM (count per million) for comparison between different samples. Log$_2$ transformed gene expression level (CPM) was used for principal component analysis (PCA) and pairwise gene expression correlation matrix analysis using R. Differentially expressed genes (DEGs) were compared with EdgeR with p value < 0.01[77]. Community detection for DEGs was done based on community detection algorithm and our improved method published in 2012[78–80]. Heatmap of modules and module expression at different samples were done by in-house script in Mathematica 11.3. Gene ontology (GO) enrichment analysis was performed using clusterProfiler package from R. GO terms with adjusted P value less than 0.05 were shown. Bubble plot was done by ggplot2 R package. All other bar plot was done using R.

TSS peaks were identified by HOMER software (default parameter) with 0.1% FDR. Peaks were annotated in the genome considering strand information with human annotation reference using HOMER. To obtain a stringent set of TSS peaks, we only considered the peaks uniquely mapped to the reference genome and the peaks supported by ≥30 reads for the following analysis. 250 bp upstream and 50 bp downstream sequences from the TSSs were extracted as the proximal promoter to perform transcription factor motif analysis by HOMER. TFs with p value < 0.01 and Benjamini q value < 0.01 were considered as significantly enriched.

## Bulk RNA sequencing

For bulk RNA-seq, total RNA concentration was calculated by Quant-IT RiboGreen (Invitrogen, #R11490). To assess the integrity of the total RNA, samples were run on the TapeStation RNA screentape (Agilent, #5067-5576). Only high-quality RNA preparations, with RIN greater than 7.0, were used for RNA library construction. Ribosomal RNA was removed before cDNA synthesis with SuperScript II reverse transcriptase (Invitrogen) and library preparation with KAPA RNA Hyper Prep Kit (Kapa Biosystems). RNA libraries were prepared according to the manufacturers' protocol (Kapa Biosystems). Libraries were pooled together for pair-end sequencing with NovaSeq 6000 or NextSeq 500 (High output) platform (illumina).

Pair-end sequencing reads of RNA-seq data were mapped to human genome (hg19 version) or mouse genome (mm10 version, for mouse bulk RNA-seq) with tophat software (tophat2, default parameter)[81], annotated repeat information was downloaded from UCSC database, repeat region quantification was done with BEDTOOLs[82] and normalized to cpm (count per million for each sample). Gene expression was calculated with cufflinks software[83]. Kolmogorov–Smirnov test (KS test) was used for comparing transposon fold-change to the background. Differential expression analysis was done using DESeq2[84].

## scRNA sequencing

H9 hESCs were transduced with control or ZBTB12 shRNA at day −3, and puromycin-selected at day −1. On the next day, Day 0 samples were kept in mTeSR1 with puromycin. For Day 8 samples, cells were cultured for 8 days in FGF2/TGF-beta deprived media (replaced every other day) containing puromycin to make sure that shRNA expression lasts throughout the differentiation. At day 0 and day 8, cells were harvested with Accutase, counted, checked for viability, and re-suspended at a concentration of 1000 cells/µl. The samples were then individually loaded on to each lane of a Chip G Chromium Controller (10× Genomics). cDNA synthesis, amplification and sequencing libraries were generated following the manufacturer's protocol. The libraries from all four samples were multiplexed together and sequenced on an Illumina NovaSeq (PE150).

The Cell Ranger toolkit (version 3.1.0) provided by 10× Genomics was applied for scRNA-seq data to align reads and generate the gene-cell unique molecular identifier (UMI) matrix, using the reference genome GRCh37. LTR7/HERVH units (named as ERV as prefix) were added in the genome annotation file using cellranger mkref command.

The expression matrices were imported into Seurat V3.1.5R package (https://satijalab.org/seurat)[85] to perform data filtration, sample integration, gene normalization, dimension reduction and data visualization. All the samples were integrated as one object by Seurat "IntegrateData" function. We excluded cells that have (1) less than 200 genes (beads with ambient RNAs), (2) more than 30,000 unique molecular identifiers (UMIs) (doublets), or (3) percent of mitochondrial reads higher than 15% (dying or stressed cells). We further excluded 576 'unidentified' cells that represent low mitochondrial genes (less than 3% of total reads) for downstream analysis. The resulting dataset consisted of 8618 cells and 26,670 genes. For dimensionality reduction and data visualization, the integrated gene expression values were log-normalized and scaled for PCA built in Seurat package[85].

Dimensionality reduction was done by Seurat "RunPCA" function with 2000 highly variable genes (default), calculating 50 principal components that capture the highest variance. Then Uniform Manifold Approximation and Projection for Dimension Reduction (UMAP) was used to visualize single-cell clusters, by graph-based clustering (resolution = 0.2) the cells finding the 20 nearest neighbors, employing the top 30 principal components with the largest variance. Cells annotated as pluripotent stem cells in Fig. 2e were subset for further analysis

(Fig. 3), and they underwent the same procedure of normalization, scaling and finding of variable genes.

For DEG analysis we used MAST algorithm[86]. Genes with $\log_2$ fold change lower than −0.25 or higher than 0.25 were further used for GO analysis (http://metascape.org). DEGs are plotted using EnhancedVolcano package with following cutoff criteria: $-\log_{10}p > 25$ and $\log_2$ fold change < −0.25 or >0.25.

For CytoTRACE[42], raw count matrix after filtering out low-quality reads is used for CytoTRACE algorithm. To generate trajectory curves using Slingshot, pluripotent stem cell cluster 2 was set as a starting cluster. In Monocle 3 trajectory analysis, 'Day 0 shCtrl' group was set to help identify root principal point.

For the comparison of the current dataset with published scRNA-seq datasets, data were integrated using Seurat integration method with 3000 features used to find anchors and 100 neighbors to filter anchors.

For weighted gene co-expression network analysis[48], we used R package WGCNA (ver 1.70) with pluripotent clusters shown in Fig. 3a. Only variable features were used to generate expression matrix for blockwise module identification. Full gene lists of three modules (turquoise, blue and brown) are provided as Supplementary Data 4.

### ChIP sequencing

ZBTB12-flag ChIP experiments were done with ChIP-IT® Express kit (Active Motif) following the protocol with anti-FLAG antibody (Cell Signaling Technologies). Then negative control input, positive control anti-PolII and anti-FLAG samples were pooled together for NextSeq 500 pair-end sequencing with 150 cycles Mid Output Kit (illumina).

For HDAC1 and H3K27ac ChIP-seq, H9 hESCs expressing shGFP (control) or shZBTB12 (clone A) were fixed with 1% formaldehyde for 15 min, quenched with 0.125 M glycine and harvested. Harvested samples were sent to Active Motif (Carlsbad, CA) for ChIP-Seq. Briefly, chromatin was isolated by adding lysis buffer, followed by disruption with a Dounce homogenizer. Lysates were sonicated and the DNA sheared to an average length of 300–500 bp with EpiShear probe sonicator (Active Motif, cat# 53051). Genomic DNA (Input) was prepared by treating aliquots of chromatin with RNase, proteinase K and heat for de-crosslinking, followed by SPRI beads clean up (Beckman Coulter) and quantitation by Clariostar (BMG Labtech). Extrapolation to the original chromatin volume allowed determination of the total chromatin yield. An aliquot of chromatin (30 μg) was precleared with protein A agarose beads (Invitrogen). Genomic DNA regions of interest were isolated using 4 μg of antibody against HDAC1 (Active Motif, Cat# 40967) or H3K27Ac (Active Motif, Cat# 39133). Complexes were washed, eluted from the beads with SDS buffer, and subjected to RNase and proteinase K treatment. Crosslinks were reversed by incubation overnight at 65 °C, and ChIP DNA was purified by phenol–chloroform extraction and ethanol precipitation. Illumina sequencing libraries (a custom type, using the same paired read adapter oligonucleotides described by Bentley et al.[87]) were prepared from the ChIP and Input DNAs on an automated system (Apollo 342, Wafergen Biosystems/Takara). After a final PCR amplification step, the resulting DNA libraries were quantified and sequenced on Illumina's NextSeq 500 (150 nt reads, paired end). Base-call data was processed and demultiplexed using bcl2fastq2 (v2.20). Pair-end sequencing reads were mapped to the human genome assembly hg19 using bwa software (version 0.7.13, default parameter)[88]. Duplicate reads were removed with samtools (version 0.1.16). The peaks were called with MACS2 software at the cutoff of 0.01 for *q* value[89].

### ZBTB12 enrichment analysis

The enrichment analysis is done by HOMER software (annotatePeaks.pl ZBTB12.peaks hg19 -gtf gencode.v19.gtf -annStats peaks_anno_stat > ZBTB12.peaks.anno). The size of ZBTB12 ChIP-seq peaks that overlap with different genomic features (SINE, LINE, LTR, etc.) was measured as a ratio (A-ratio). The size of each genomic feature over the human genome was used as a background ratio (B-ratio). The enrichment score was calculated by $\log_2(A/B)$.

### ZBTB12 conservation analysis

ZBTB12 protein sequence of different species (including human, chimpanzee, macaque, mouse, rat, dog, and zebrafish) were retrieved from NCBI database. Multi sequence alignment were done using MEGA X[90]. Neighbor joining tree[91] was constructed using MEGA X. ZBTB12 protein sequences were aligned using MEGA X and edited by Genedoc[92].

### Divergence time analysis

A 940 Full LTR unit (two LTR7 flanking HERVH along a LTR unit) were used for estimating divergence time analysis. Sequence of two LTR7 flanking LTR7/HERVH unit were retrieved from human genome sequence (hg19 version download from UCSC database), paired LTR were aligned with MUSCLE[93]. Then divergence time was calculated by in house code using Kimura 2 parameter model (K2P model: Kimura 2 parameter correction $K = 1/2ln[1/(1 − 2P\text{-}Q)] + 1/4ln[1/(1\text{-}2Q)]$, human evolution rate $r = 1.28e−8$)[94].

### Reporting summary

Further information on research design is available in the Nature Portfolio Reporting Summary linked to this article.

## Data availability

The ChIP-seq and single cell transcriptomics data generated in this study have been deposited in the Gene Expression Omnibus database (https://www.ncbi.nlm.nih.gov/geo/) under accession codes GSE118946, GSE167052, and GSE205342. NANOG ChIP-seq data were downloaded from the database GSM2816625. Source data are provided with this paper.

## Code availability

Code-based analysis is done following the standard guidelines of the packages. User-defined parameters are provided in the Methods section. No custom code is developed or used. Analysis scripts will be made available upon request.

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

## Acknowledgements

This work was supported by Larry L. Hillblom Foundation (2018-A-0004-NET), a NIH grant (U54NS100717), the National Research Foundation of Korea (NRF) grant funded by the Korea government (MSIT) (NRF-2019R1C1C1002377, NRF-2020M3A9D8038184, NRF-2021R1A4A1031754, NRF-2022R1F1A1063619), and the BK21 FOUR. We thank Hyuk-Jin Cha at Seoul National University for providing J1 mEpiSCs and Clara Kim at UC Santa Barbara for constructing plasmids that helped this work.

## Author contributions

Conceptualization: D.H., G.L., Y.O., K.S.K., and J.J.; Methodology: D.H., G.L., and Y.O.; Validation: S.O., S.Y., and L.M.; Formal analysis: D.H., G.L., and M.C.A.; Investigation: D.H., G.L., Y.O., S.O., S.Y., L.M., N.R., and M.C.A.; Writing-Original Draft: D.H., G.L., K.S.K., and J.J.; Writing-Review & Editing: D.H., G.L., Y.O., S.O., S.Y., K.S.K., and J.J.; Supervision: K.S.K. and J.J., Funding acquisition: K.S.K. and J.J.

## Competing interests

The authors declare no competing interests.
