## [Peer Review File · Nature Communications]

ZBTB12 is a molecular barrier to dedifferentiation in human pluripotent stem cellsReviewer #1 (Remarks to the Author):

Han et al.

In this study Han et al. report on the role of the transcription factor ZBTB12 in the regulation of human pluripotent stem cells (hPSC). Using CAGE sequencing to identify active transcription start sites, the authors found that many promoter regions in undifferentiated hPSC have binding sites for this transcription factor. Knockdown studies revealed that reduction in ZBTB12 led to an increase in expression of some genes characteristic of the naïve state, increased self-renewal capacity, and decreased response to two differentiation protocols. scRNA-seq showed that cells deficient in ZBTB12 failed to differentiate and reverted instead towards a state that more nearly resembled naïve pluripotency. Further investigation showed that ZBTB12 negatively regulates LTR7/HERVH expression, and that downregulation of several lncRNAs controlled by ZBTB12 is required for early stages of germ layer differentiation.

The molecular mechanisms that regulate transitions of pluripotent cells along the spectrum from the naïve state to primed pluripotency and then to germ layer specification are the topic of much current investigation. This study identifies a new transcription factor that appears to be critical to the entry of primed pluripotent stem cells into lineage specification, a potentially exciting finding. The study would be strengthened by additional bioinformatics analysis to place the ZBTB12 knockdown cells in the context of different pluripotent states, for example through more detailed comparisons with naïve and primed hPSC datasets, and datasets from cultured pre- and post- implantation human and monkey embryos available in the literature. As it stands, the precise developmental status of these cells is unknown, and it is possible that they may have no existing counterpart. Also, a number of recent reports have described human cells at intermediate stages between naïve and primed pluripotency, and it would be important to understand where the knockdown cells align on this spectrum. The authors do not mention formative pluripotency explicitly, but it is possible that ZBTB12 might be critical to the process of capacitation, whereby naïve cells acquire competence to undergo lineage differentiation. Reference 11 describes a subset of hPSC that express high levels of HERVH, and again, it would be useful to understand how the authors' study compares with this previous work. The mechanistic links between HERVH, lncRNA targets, and differentiation, which comprise much of the study, could be stronger.

Specific points:

- 1. Page 2-the authors should discuss the concept of formative pluripotency, and recent studies that have described intermediate states between naïve and primed pluripotency, including work from the Studer, Wu, Pera, Bhatia, and Smith laboratories.**
- 2. Page 3-under what circumstances would primed hPSC de-differentiate, except when induced to do so experimentally?**
- 3. Page 3-high expression of HERVH in the blastocyst is not really compatible with its presence in primed hPSC; see 4 below**
- 4. Page 4-reconcile expression of HERVH in primed cells with this postulated role in naïve cells. The authors favor the concept of this molecule serving as a rheostat, which is certainly an interesting idea, but this is not articulated or investigated all that directly here. See also above discussion regarding Ref. 11.**
- 5. Page 7-does the speckled nucleoplasmic staining with ZBTB12 tell us anything about likely function of this gene?**
- 6. Page 7-gene expression in the knockdown cells grown in maintenance conditions should be examined; the increased colony formation represents a marked change in phenotype**
- 7. Page 7-what findings were confirmed in cell lines other than WA09**
- 8. Page 7 and Supp Figure 2g and following results sections-ZBTB12 overexpression is massive at the RNA and presumably protein levels, it is questionable how meaningful these experiments are.**
- 9. Page 7-did the authors examine other differentiation paradigms apart from growth factor withdrawal or EB formation, i.e. did they look at induction of differentiation by**

morphogens?

10. Page 8-as noted above the comparison of control and KD cells in the undifferentiated state is critical. How long was the shRNA on in undifferentiated cells before this assay? More detail here would be helpful. The analysis here suggests that the key effect of knockdown was to suppress spontaneous differentiation or lineage priming. This would accord with an action of holding the cells in a formative state.

11. Page 9-data suggest that KD does not prevent a very substantial loss of colony forming ability relative to that shown in 1c. Obviously the cell state has changed in some significant way, this does not indicate functional preservation of self-renewal. Do cells expressing shRNA maintain a capacity to differentiate once expression stops?

Conditional shRNA expression would be more informative here.

12. Page 10-a much more extensive bioinformatics analysis, as noted above in general comments, would be required to strengthen any conclusions regarding the developmental status of the KD cells. The data in 3G are not that compelling in this regard. Overexpression studies are not that informative for reasons noted above.

13. Page 12 Figure 4f-h-actually the changes in transcripts for HERVH or LTR7 are modest; eventually they decline to control levels during differentiation. This contrasts with the data on stem cell gene expression which does not seem to change globally in KD cells?

14. Page 13-what are the authors trying to assert here? There is no direct evidence to show a particular interaction between NANOG and ZBTB12 in regulation of LTR7/HERVH.

15. Page 16-eventually the LINC-ROR declines to control levels in the study in 6d, even though the state of the cells is quite different.

16. Figure 6f-isolated fields showing a few cells are not convincing, the authors should carry out flow cytometry or some quantitative analysis. Same comment with Figure 6k.

Reviewer #2 (Remarks to the Author):

In the manuscript entitled "ZBTB12-HERVH-LncRNA axis is a molecular barrier for dedifferentiation of human stem cells", Han et al claim that: 1. Transcription repressor ZBTB12 is required for the differentiation of human pluripotent stem cells (hPSCs) into three germ layers by acting as a molecular barrier for the dedifferentiation of hPSCs to more primitive naive like state. 2. ZBTB12 repress the expression of HERVH, especially newly emerged and active ones. 3. ZBTB12 functions through HERVH overlapping lncRNAs including ESRG and LINC-ROR. These findings are interesting and novel, should be of interest to scientists in fields of stem cells, gene regulation and retrotransposon regulation and evolution. In particular, their findings may potentially lead to the forming of new concept on the domestication of ERVs. Their claims are generally supported by their data.

My major concerns are:

1. The claim on ZBTB12 acting as the barrier for the dedifferentiation of hPSCs needs to be tested in different differentiation conditions. For example, during embryoid body differentiation, neural differentiation or other directed differentiation conditions, do authors also see the upregulation of naive markers upon ZBTB12 knockdown?

2. The function of LINC-ROR and ESRG in the dedifferentiation process needs to be evaluated. Specifically, under spontaneous differentiation condition, whether overexpressing LINC-ROR or ESRG leads to the upregulation of naive state genes?

3. The authors claim that the function of ZBTB12 in blocking dedifferentiation is newly evolved in primate. This claim needs further evidence. What happens if knocking Zbtb12 in mouse epiblast stem cells? Answering this question may also help to clarify the function of human specific lncRNAs in dedifferentiation.

My minor concerns include:

1. The title may need to be rephrased if further evidence is not provided. First, the role of HERVH and lncRNA in the dedifferentiation is not defined; second, human stem cells should human pluripotent stem cells.
2. Page 11, "Collectively, these results suggest that ZBTB12 serves as a molecular barrier for a naïve-like pluripotent state and this role of ZBTB12 is essential for successful differentiation of hESCs."; "is essential" better to be "may be".
3. The cooperative model of ZBTB12 and NANOG in regulating HERVH expression needs to be further clarified. Do ZBTB12 and NANOG antagonize each other in binding HERVH sites? Or do they bind together?
4. Figure 4f is not consistent with Figure 4h. It could be that authors used different normalization methods. By the way, how data are normalized should be described in figure legends. For example, Data were normalized to Day 0 shControl.
5. Figure 5a, a scale for the peak file will help to judge how far ZBTB12 binding site is away from LINC-ROR TSS.
6. Figure 5f, pLV_VPA samples should be shown. If the expression of LTR7 et al is significantly upregulated by VPA, the claim on ZBTB12 functioning through recruiting HDAC is flawed. If ZBTB12 works through HDAC pathway, after knocking down ZBTB12 in hESCs, VPA treatment should have minimum impact on the expression of LTR7, unless there exists other mechanism to recruit HDAC to LTR7 locus.
7. The crosstalk between ZBTB12 and SIN3A/HDAC complex needs to be further clarified. ChIP using antibodies to SIN3A/HDAC in ZBTB12 knocking down cells may help to build a model that ZBTB12 recruits SIN3A/HDAC to repress HERVH.
8. For experiments in Figure 1e, 1h, 6g, 6k, it would be nice to have more quantitative data such as qPCR or Western.
9. Supplementary Figure 7h, the data for shA and shB are not consistent, see the expression of NANOG in shControl.
10. Is ZBTB12 downregulated in naive hPSCs and upregulated during differentiation?
11. Data point from same batch of experiment may be labelled by the same color or same shape.
12. Statistic test should not be made for $n = 2$ (e.g. Supplementary Figure 7d).
13. For multiple comparison, appropriate statistic tests should be applied. Current version used Student's t-test throughout the manuscript.

Reviewer #3 (Remarks to the Author):

Han et al are going after a highly interesting subject: regulation of human pluripotency. Their pipeline reveals ZBTB12 as a novel candidate that controls the exit from a naïve/naïve-like pluripotency, thus removes a barrier from cell differentiation. Their single cell RNA sequencing reveals that ZBTB12 was essential for three germ layer differentiation by blocking dedifferentiation of hPSCs toward a more primitive state. Mechanistically, they suggest that ZBTB12 serves as a master repressor of human endogenous retrovirus H (HERVH), previously implicated in regulating pluripotency in primates. The authors claim at identifying the ZBTB12-HERVH-lncRNA axis as a molecular machinery that safeguards the unidirectional transition of metastable stem cell fates toward developmentally advanced states. The story is highly interesting, but

only partially supported.

To identify novel regulators of pluripotency, the authors use a strategy that is not dependent on expression changes of TFs, and could be used to precisely map transcription start sites (TSSs). This approach identifies ZBTB12. ZBTB12 is a phylogenetically conserved protein (here, shown between mice and human), and has a broad expression in multiple cell types and tissues. Notably, its role in regulating pluripotency is not known.

The first part of the manuscript is relatively solid. The authors convincingly demonstrate that ZBTB12 is a novel gene that controls exit from naivety.

Importantly, phylogenetically, HERVK is much younger than HERVH. HERVH and HERVK cannot be mentioned together, because they are not regulated the same way and they are marking different cell types. Notably, the study seems to reveal a different type of naivety upon KD ZBTB2, analyzed by single cell RNA-seq, or when they use the LIF, 3i protocol to induce a naive state. In the second case, HERVK is upregulated much stronger than TFCP2L1. What about the HERVH-related products in this condition? Are these different types of cells? Recent studies suggest that HERVK is upregulated in artificially induced naivety (where HERVH is not highly expressed), while high HERVH expression marks a subpopulation of naive-like cells, naturally present in PSC cultures. The relationship between HERVK vs HERVH should be clarified in the manuscript.

The second part, where the authors try to decipher the underlying mechanism, is far less convincing. They claim that "As a downstream mechanism, ZBTB12 binds to and transcriptionally suppresses HERVH." However, this is not well supported by their data.

The study supports the previously suggested role of HERVH and HERVH-derived transcripts/lncRNA in regulating naive-like pluripotency. The control of the HERVH-mediated pluripotency regulation is not fully understood. While, the set of TFs, whose expression upregulates HERVH transcription in pluripotent stem cells is reported, the suppression of the regulatory network is yet to be deciphered. Because ZBTB12 is expressed antagonistically with HERVH, a HERVH-derived gene ESRG and LINC-ROR, the authors focused on this relationship. They present evidence that upon depletion of ZBTB12 RNA, pluripotency is affected. However, whether this effect can be linked to a direct control over HERVH by ZBTB12, is not supported well.

"Significant correlation of HERVH expression with the overlapping lncRNAs was observed after ZBTB12 KD (Pearson correlation analysis, $R=0.53$, $P<0.001$), suggesting that ZBTB12 regulates expression of the lncRNAs through LTR7/HERVH (Fig. 6a, Supplementary Table 6). - These data present correlation, but not a direct evidence.

Although ZBTB12 binds DNA (ChIP-seq), this binding does not seem to be specific to HERVH sequences (>200 copies upregulated in pluripotency). In fact, the binding of ZBTB12 on HERVH elements is not significant: The authors find that "...ZBTB12 binding sites were frequently found in up- or downstream of HERVH loci." ; "... among 70 ZBTB12 ChIP-seq peaks associated to full length HERVH loci (Supplementary Table 7), only three peaks were located on LTR7."

Consider, if we randomly distribute ChIP-seq peaks on human genome, it is very likely that ~10-20 HERVH loci would be occupied by chance. Thus, it is not statistically significant that ZBTB12 binds and regulates HERVH-derived sequences and pluripotency.

Let's take the possibility that ZBTB12 is specifically binding on three copies of HERVH. Even the identified 3 peaks map to different parts of HERVH sequences, arguing against phylogenetic conservation. Is it possible that ZBTB12 controls only these copies, and these are the key regulators? Notably, the study is not following up on these particular copies, but on ESRG and LINC-ROR. Indeed, the HERVH products (ESRG, LINC-ROR) were already shown by others to effect pluripotency.

Overall, it is not clear what might provide the specificity of the putative ZBTB12-mediated HERVH (if any) regulation. Their hypothesis that ZBTB12 controls HERVH via recruiting SIN3A/HDAC1 complex is interesting, but not convincing either. The study demonstrates an immunoprecipitation of ZBTB12-SIN3A and ZBTB12-HDAC1. This might suggest that similar to ZBTB16 (PLZF), ZBTB12 is involved in a protein-protein interaction with the SIN3A/HDAC1 complex. The hypothesis that ZBTB12 works in conjunction with the SIN3A/HDAC complex is likely, but again the data does not support direct connection with HERVH regulation. A dissection of a complex, bound to a HERVH locus could identify the components. This approach might substantiate that SIN3A/HDAC is recruited to a specific HERVH locus by ZBTB12.

The evolutionary aspect might be interesting. The study shows that the mouse ZBTB12 is able to inhibit HERVH expression, however it might be interesting to find out whether ZBTB12 also controls the exit from naivety in mice. Is the proposed regulatory function of ZBTB12 specific to human/primates?

The sequence of the ZBTB12 is conserved between mouse and human. Are the aa changes under selection? Does ZBTB12 binds differentially to genomic DNA in mice?

Further issues:

"... mouse PSCs cultured with serum exhibit substantial intrapopulation heterogeneity with a mixture of naïve- and primed-like cells, which represents a dynamic equilibrium of cells transitioning between distinct metastable states^{9,10}."

- Importantly, human PSC cultures are also heterogenous, however they have much lower percentage of naive-like cells.

"ZBTB12-mediated suppression drives efficient shutdown of young and transcriptionally active HERVH loci upon pluripotency exit. These results suggest that ZBTB12 is engaged in regulating young and transcriptionally active HERVH loci."

- Nota that young is not equal to transcriptionally active! The HERVH cannot be considered as a young element. Perhaps, the authors mean less diverged?

Figure 3g.

"Treating hESCs with LIF and 3i was sufficient to transiently activate the expression of naïve pluripotency-related genes and ZBTB12 KD further promoted naïve gene activation (Fig. 3g)."

- Show ESRG and LNC-ROR on the figure. Curiously, HERVK is much more induced than TFCP2L1 or NANOG..!

"Despite the high expression level in hESCs, ESRG KD did not have any significant effects on the expression of core pluripotency genes (Supplementary Fig. 7f, g)."

- Confusing! The cited figures show significance.

"Focusing on LTR7/HERVH, we identified 280 lncRNAs overlapped with LTR7/HERVH loci in the human genome (Supplementary Table 8). Among them, 56 lncRNAs were detected in our RNA-seq data with counts per million (CPM) higher than one (Supplementary Table 9)". - Similar data has been already reported in¹¹.

Reviewer #4 (Remarks to the Author):

The authors identify ZBTB12 as pluripotent stem cell associated transcription factor. They find that ZBTB12 knock-down impaire exit from pluripotency and ZBTB12 overexpression allows better differentiation. Using ChIP-seq of a tagged ZBTB12

protein, they identify genome-wide binding sites. Interestingly, numerous binding sites appear associated with HERV sequences. ZBTB12 knock-down results in higher HERV expression (with focus on LTR7 driven HERVs), suggesting a repressive role of ZBTB12. Interestingly, there is an inverse correlation between Nanog knock-down and ZBTB12 knock-down on HERV expression, suggesting that ZBTB12 counteracts Nanog activation of HERVs. The authors find a partial overlap between ZBTB12 peaks and HDAC/SIN3A binding sites, suggesting a mechanism for transcriptional repression. Furthermore ZBTB12 interacts with HDAC/SIN3A. Finally, the authors study HERV-associated lncRNAs and their role in pluripotency exit. They show that two lncRNAs, LINC-ROR and ESGR influence pluripotency exit. The authors propose that ZBTB12 represses HERV sequences and HERV associated lncRNAs. Loss of ZBTB12 results in HERV and lncRNA derepression, which somehow impairs pluripotency exit.

In its current state, the manuscript suggests an important role of ZBTB12 in human pluripotent stem cells. However, the mechanism is very loosely established. The role of ZBTB12 in HERV repression is not fully clear. There are ZBTB12 peaks in some distance to LTR7/HERV sequences, however, how this binding affects HERV transcription at a distance is unclear. Also, the transcriptional regulation of HERV-associated lncRNAs is not well described. Are these lncRNAs actually all driven from LTR7 promoters, as suggested in the screenshot for LINC-ROR? This would rather indicate that HERVs have evolved into regulatory sequences of these lncRNAs and the regulation by ZBTB12 is not so much towards HERVs, but rather towards specific transcripts which by chance utilize LTR promoters. In this sense the title suggesting a ZBTB12-HERV-lncRNA axis is a bit misleading. Based on the authors demonstrating the effect of two lncRNAs on pluripotency, the potential mechanism of action of these lncRNAs should be discussed more extensively.

Specific comments:

(1) In order to connect ZBTB12 with transcriptional regulation a clear intersection between ZBTB12 binding sites and transcripts in the vicinity which change expression in ZBTB12 knock-down should be established.

In the initial screen the authors identify ZBTB12 as potential transcription factor binding to TSS of the nanoCAGE transcripts. Could ZBTB12 binding to these sequences be confirmed in the ChIP-seq data?

(2) The ZBTB12 ChIP-seq data need to be quality controlled better before they can be judged. The identified binding motif needs to be compared to the reference motif. The full result table of the de novo motif finding analysis needs to be provided. This also helps to judge the co-enriched motifs.

The binding sites (and genomic features on binding sites) should be compared to the published profile in HEK cells.

(3) Binding of ZBTB12 to HERV sequences needs to be more specifically investigated and presented. The example screenshot in Fig 4e demonstrates binding to an LTR7 promoter. However, only 7 peaks associate with LTR7 promoters, indicating that direct HERV binding is strongly under-represented. Later in the manuscript the authors come back to this point and argue that there are cis effects of distant ZBTB12 binding. This relates to point (1) which genomic features are associated with ZBTB12 mediated transcriptional effects. Their final argument is that ZBTB12 regulates lncRNAs LINC-ROR and ESGR, so at least for these two examples, ZBTB12 binding and transcriptional effects should be shown in screenshots.

(3) Transcriptional changes in HERVs should be better represented on the family level or on the level of individual transcripts (especially if there is a focus on specific lncRNAs) with baseMean vs log2fc plots. Is there a transcriptional change on whole HERV families or is there effects on specific HERV integrations? Which HERVs are bound by ZBTB12 (or indirectly affected in cis)? Transcriptional changes in LTR7 in Fig 4f,g: which copies are detected with these primers?

(4) Repression by ZBTB12 is suggested to be mediated by HDAC/SIN3A. Fig.5b showing co-occupancy between these factors and ZBTB12 clearly shows only partial overlap between these factors. To analyze these data more robustly, an overlap analysis (and k-means clustering) needs to be performed to identify ZBTB12 peaks which associate with HDAC and SIN3A (show example screenshots). These peaks should be intersected with ZBTB12 regulated transcripts. If there is a good overlap between ZBTB12 repressed transcripts and HDAC/Sin3A binding, it should be tested if recruitment of HDAC/SIN3A depends on ZBTB12 (and vice versa). Treatment of cells with HDAC inhibitors as done by the authors, is known to up-regulate HERVs and is not specific to ZBTB12 activity.

(5) most figures are hard to read, please increase font size in all figures

We appreciate all of the reviewers for their critical and constructive comments that significantly improved our work. The revised manuscript clearly demonstrates the core novel finding of a molecular barrier to dedifferentiation. We agree some of the criticisms concerning the mechanism of ZBTB12 control over HERVH and have therefore presented our data with more conservative claims.

REVIEWER COMMENTS

Reviewer #1 (Remarks to the Author):

Han et al.

In this study Han et al. report on the role of the transcription factor ZBTB12 in the regulation of human pluripotent stem cells (hPSC). Using CAGE sequencing to identify active transcription start sites, the authors found that many promoter regions in undifferentiated hPSC have binding sites for this transcription factor. Knockdown studies revealed that reduction in ZBTB12 led to an increase in expression of some genes characteristic of the naïve state, increased self-renewal capacity, and decreased response to two differentiation protocols. scRNA-seq showed that cells deficient in ZBTB12 failed to differentiate and reverted instead towards a state that more nearly resembled naïve pluripotency. Further investigation showed that ZBTB12 negatively regulates LTR7/HERVH expression, and that downregulation of several lncRNAs controlled by ZBTB12 is required for early stages of germ layer differentiation.

The molecular mechanisms that regulate transitions of pluripotent cells along the spectrum from the naïve state to primed pluripotency and then to germ layer specification are the topic of much current investigation. This study identifies a new transcription factor that appears to be critical to the entry of primed pluripotent stem cells into lineage specification, a potentially exciting finding.

A: We appreciate the reviewer for finding our study exciting.

The study would be strengthened by additional bioinformatics analysis to place the ZBTB12 knockdown cells in the context of different pluripotent states, for example through more detailed comparisons with naïve and primed hPSC datasets, and datasets from cultured pre- and post- implantation human and monkey embryos available in the literature. As it stands, the precise developmental status of these cells is unknown, and it is possible that they may have no existing counterpart. Also, a number of recent reports have described human cells at intermediate stages between naïve and primed pluripotency, and it would be important to understand where the knockdown cells align on this spectrum.

A: As suggested, we performed detailed bioinformatic analyses with published scRNA-seq from human pre- and post-implantation embryos and naïve and primed hESCs. Our results showed that Day8 shZBTB12 cells clustered closely to human pre-implantation epiblasts and naïve hESCs. Please see our response to Comment 12.

The authors do not mention formative pluripotency explicitly, but it is possible that ZBTB12 might be critical to the process of capacitation, whereby naïve cells acquire competence to undergo lineage differentiation.

A: We mention formative pluripotency in the Introduction of the revised manuscript. Based on the expression of formative signature genes in our scRNA-seq data, we found that shZBTB12 cells did not show a transcriptomic profile of formative hPSCs. Please see our response to Comment 10.

Reference 11 describes a subset of hPSC that express high levels of HERVH, and again, it would be useful to understand how the authors' study compares with this previous work.

A: The elevated expression of HERVH and dedifferentiation to a naïve-like state by ZBTB12 KD are consistent with the findings in Reference 11 (Ref 16 in the revised manuscript). As mentioned above, extensive bioinformatic analyses with published scRNA-seq data confirmed that ZBTB12 KD drives primed hESCs towards a naïve-like state upon differentiation.

The mechanistic links between HERVH, lncRNA targets, and differentiation, which comprise much of the study, could be stronger.

A: See the specific points below.

Specific points:

1. Page 2-the authors should discuss the concept of formative pluripotency, and recent studies that have described intermediate states between naïve and primed pluripotency, including work from the Studer, Wu, Pera, Bhatia, and Smith laboratories.

A: As suggested, formative pluripotency is now discussed in the Introduction. Please see Page 2 Line 23.

2. Page 3-under what circumstances would primed hPSC de-differentiate, except when induced to do so experimentally?

A: The known presence of a naïve-like population in primed hPSC culture suggests that hPSCs can undergo dynamic cell state transitions such as dedifferentiation to a naïve-like state and differentiation back to a primed state.

Accordingly, it was also reported that a small proportion of 8C-like totipotent cells exists in naïve hPSC culture (Taubenschmid-Stowers et al., Cell Stem Cell, 2022, doi.org/10.1016/j.stem.2022.01.014). Although dedifferentiated naïve-like cells are rare in primed hPSCs, ZBTB12 KD in the absence of FGF2 and TGF- β (two strong primed state-inducing factors) showed robust dedifferentiation of primed hPSCs toward a naïve-like state, highlighting the significance of dedifferentiation barriers in pluripotent states even in the context of well-recognized metastable pluripotent states.

3. Page 3-high expression of HERVH in the blastocyst is not really compatible with its presence in primed hPSC; see 4 below

A: See our response to Comment 4.

4. Page 4-reconcile expression of HERVH in primed cells with this postulated role in naïve cells. The authors favor the concept of this molecule serving as a rheostat, which is certainly an interesting idea, but this is not articulated or investigated all that directly here. See also above discussion regarding Ref. 11.

A: In the revised manuscript, we provide data showing HERVH expression in naïve-like, primed, and differentiated states (Supplementary Fig. 5m). Although HERVH expression is higher in naïve-like cells than primed cells, primed cells still show significantly elevated HERVH expression compared to differentiated cells. In qPCR, the raw Ct values of HERVH are 11-13 in primed hESCs while that of GAPDH is 15-17. Given the gradual decline of HERVH expression during the pluripotent state transition together with its role in pluripotent states (Wang et al, 2014, Lu et al, 2014 and our data), HERVH is proposed as a molecular rheostat for dynamic pluripotent states.

5. Page 7-does the speckled nucleoplasmic staining with ZBTB12 tell us anything about likely function of this gene? –

A: Recent studies have suggested that transcription factors undergo liquid-liquid phase separation to exert their activities (Hnisz et al., 2017). One possible idea is that ZBTB12 functions in a phase-separated compartment, which would be an interesting subject for future research.

6. Page 7-gene expression in the knockdown cells grown in maintenance conditions should be examined; the increased colony formation represents a marked change in phenotype

A: As suggested, we analyzed the differentially expressed genes (DEGs, using MAST algorithm) in day 0 samples of scRNA-seq data. Despite the dramatic change in LTR expression (Fig. 4c and d), ZBTB12 KD showed mild transcriptional alteration in protein coding genes with 64 up-regulated ($\log_2FC > 0.25$) and 130 down-regulated ($\log_2FC < -0.25$) DEGs. Down-regulated DEGs include several developmental genes related to FGF, BMP, and WNT signaling pathways (FGFR1, ID1, ID2, ID4, SFRP1). Gene ontology (GO) analysis using metascape revealed that the down-regulated DEGs are associated with GO terms related to stem cell differentiation (GO:0030855-epithelial cell differentiation, GO:0048729-tissue morphogenesis, GO:0060322 head development). GO analysis with up-regulated DEGs showed cholesterol metabolism with the most significant p value. Activated lipid metabolism was recently reported to be a key feature of naïve pluripotency (Cornacchia et al., 2019, Cell Stem Cell 25, 120–136). Overall, these results support our observation that ZBTB12 KD drives hESCs to a self-renewing state away from differentiation. These results are described in the revised manuscript (Supplementary Fig. 3g, h). Please see Page 8 Line 21.

7. Page 7-what findings were confirmed in cell lines other than WA09

A: In the original manuscript, we confirmed major findings below in H1 and IMR90-iPSC lines

- 1) H1 hESCs- Effect of ZBTB12 KD on colony-initiating ability (Fig 1c)
- 2) H1 hESCs- Nuclear expression of endogenous ZBTB12 proteins (Supplementary fig 2a)
- 3) H1 hESCs and IMR90-iPSCs- Effect of ZBTB12 KD on HERVH expression (Supplementary fig 5f, g)

In the revised manuscript, we further tested the effect of ZBTB12 KD on differentiation with H1 hESCs. Please see Supplementary Fig. 2k.

8. Page 7 and Supp Figure 2g and following results sections-ZBTB12 overexpression is massive at the RNA and presumably protein levels, it is questionable how meaningful these experiments are.

A: While mRNA level was almost ~92 fold higher when ZBTB12 was overexpressed, ZBTB12 protein was only ~6.7 fold induced, measured by immunofluorescence assay (data added in Supplementary Fig 2h). Still, we agree that overexpression definitely has limitations. However, overexpression of the gene of interest has been widely used as a gain-of-function method, which, as we done here, supports the results from loss-of-function experiments. In addition, overexpression can be used for rescue experiments as we did in Fig. 1h where the phenotype of ZBTB12 knockdown was reversed by ZBTB12 overexpression.

9. Page 7-did the authors examine other differentiation paradigms apart from growth factor withdrawal or EB formation, i.e. did they look at induction of differentiation by morphogens?

A: As suggested, we conducted neuroectoderm-direct differentiation using a robust dual SMAD Δ i protocol. Consistent with the effect in spontaneous differentiation, ZBTB12 KD inhibited neuroectoderm differentiation, assessed by qPCR

and Western blotting. These results are shown in Supplementary Fig. 2m in the revised manuscript. Please see Page 7 Line 22.

10. Page 8-as noted above the comparison of control and KD cells in the undifferentiated state is critical.

A: Please see our response to Comment 6.

How long was the shRNA on in undifferentiated cells before this assay? More detail here would be helpful.

A: We transduced shRNA lentivirus at day -3, treated with puromycin at day -1 for 24h to select the cells expressing shRNA and harvested Day 0 samples for single cell RNA sequencing library prep. For day 8 samples, cells were cultured for 8 days in FGF2/TGF-beta deprived media (replaced every other day) containing puromycin to make sure that shRNA expression lasts throughout the differentiation. Cells were then harvested and processed to scRNA-seq. The detailed experimental procedure has been added in Method.

The analysis here suggests that the key effect of knockdown was to suppress spontaneous differentiation or lineage priming. This would agree with an action of holding the cells in a formative state.

A: We performed gene set enrichment analysis with a signature gene set for a human formative state and found no difference between undifferentiated control and ZBTB12 KD cells, which is further confirmed by key marker genes for naïve, formative, and primed states. These results are in part due to the robust primed culture condition, mTeSR1, that strongly hold hESCs in a primed state. These results are shown in Supplementary Fig. 4b-d in the revised manuscript. Please see Page 11 Line 5.

11. Page 9-data suggest that KD does not prevent a very substantial loss of colony forming ability relative to that shown in 1c. Obviously the cell state has changed in some significant way, this does not indicate functional preservation of self-renewal. Do cells expressing shRNA maintain a capacity to differentiate once expression stops? Conditional shRNA expression would be more informative here.

A: As suggested, we used IPTG-inducible shRNA vectors to knockdown ZBTB12. In H9 hESCs, IPTG treatment induced ZBTB12 downregulation, which was recovered upon IPTG withdrawal. As expected, H9 hESCs cultured and differentiated with IPTG showed significant upregulation of OCT4 and NANOG in ZBTB12 KD cells after spontaneous differentiation. The mild effect is probably due to less efficient ZBTB12 KD in the IPTG-inducible system. However, IPTG withdrawal reverted the effect, suggesting that ZBTB12 KD-mediated pluripotency exit delay is a reversible phenotype. These results are shown in Supplementary Fig. 2n-p in the revised manuscript. Please see Page 8 Line 2.

12. Page 10-a much more extensive bioinformatics analysis, as noted above in general comments, would be required to strengthen any conclusions regarding the developmental status of the KD cells. The data in 3G are not that compelling in this regard. Overexpression studies are not that informative for reasons noted above.

A: We performed a much more extensive bioinformatic analysis. We compared our scRNA-seq data with three published scRNA-seq datasets: *Petropoulos et al. (2016, pre-implantation stage E3-E7)*, *Messmer et al. (2019, naïve and primed ESCs)* and *Tyser et al. (2021, Epiblasts from a gastrulating human embryo)* (Fig. 3f, g). All datasets were integrated and projected on a single UMAP space. Most cells from pre-implantation human embryos (Petropoulos) were clustered together with naïve hESCs (Messmer). Epiblasts cells (Tyser) were located across naïve and primed clusters with a slight enrichment in primed clusters (Messmer), which indicates transitioning from a naïve to a primed pluripotent state (Fig. 3f, g). These combined references show that clustering and UMAP algorithm reasonably define and visualize cell states.

When our dataset (the Seurat object shown in Fig 3a) was plotted with and compared to the references (Fig. 3g), Day 0 undifferentiated cells were clustered closely to the primed cells from Messmer et al., suggesting these cells are mostly in a primed state as expected. In stark contrast, Day 8 shZBTB12 cells are closely located to the naïve (Messmer et al.), E5 and E6 pre-implantation (Petropoulos et al.) cells, supporting our hypothesis of naïve-like status. Day 8 shCtrl cells were located distinctly from naïve or primed (Messmer et al.) and closely to E7 (Petropoulos et al.) cells. When we defined new clusters (cluster A to D) using all-combined dataset (Fig. 3h), Day 8 shCtrl cells (cluster C) clustered distinctly from naïve-, pre-implantation, and Day 8 shZBTB12 clusters (cluster A). Day 0 cells were clustered together with the major population of primed (Messmer et al.) cells (Fig. 3h). These results are in accord with our original conclusion.

Furthermore, we applied weighted gene co-expression network analysis (WGCNA) to our pluripotent stem cell dataset (Fig 3a) to identify unique gene modules in a completely unbiased way (Supplementary Fig. 4e-h). Cluster dendrogram identified three gene correlation network modules (brown, turquoise and blue) (Supplementary Fig. 4e). Cluster 1 where Day 8 shZBTB12 cells are predominantly enriched has the highest blue module eigengene score compared to Cluster 0 and 2, indicating that the genes in the blue module either characterize or correlate with the cell state of Cluster 1 (Supplementary Fig. 4f, g). Surprisingly, many ERV elements (including LINC-ROR), naïve marker genes (NANOG, DPPA5 and UTF1) and genes found in naïve gene set (UTF1, DPPA5, MT1G, MT1H and NANOG) (Liu et al., 2020) were found in the blue module (Supplementary Fig. 4h).

Altogether, these data strongly support the idea of dedifferentiation into a naïve-like state in ZBTB12 KD cells after 8 days of FGF2/TGF-beta deprivation. These data are shown in Fig. 3f-h, Supplementary Fig. 4e-h and Supplementary Table 5 in the revised manuscript. Please see Page 11 Line 8.

13. Page 12 Figure 4f-h-actually the changes in transcripts for HERVH or LTR7 are modest; eventually they decline to control levels during differentiation. This contrasts with the data on stem cell gene expression which does not seem to change globally in KD cells?

A: The primers for LTR7, HERVH GAG and HERVH POL amplify multiple targets (49, 423 and 320 respectively, now provided as Supplementary table 8). In qPCR, the raw Ct values of these targets are 11-13 while that of GAPDH is 15-17, indicating high expression levels of LTR7/HERVH transcripts. In this context, we think that 20 to 50% increase by ZBTB12 KD is not modest. Furthermore, our bulk RNA-seq data showed significant upregulation of LTR7/HERVH in ZBTB12 KD cells (Fig. 4d).

In the early stages of differentiation from Day0 to Day6, ZBTB12 KD cells show significantly higher LTR7/HERVH expression than control cells, which contributes to delayed pluripotent exit. As the reviewer pointed out, the elevated LTR7/HERVH expression in ZBTB12 KD cells declines to that of control cells at Day 10. These results suggest that pluripotency exit is delayed, but not completely blocked in ZBTB12 KD cells. Maintenance of OCT4 and NANOG expression in ZBTB12 KD cells in Fig. 1e and 1h was observed in day 6 after differentiation, in which HERVH expression also stays high.

14. Page 13-what are the authors trying to assert here? There is no direct evidence to show a particular interaction between NANOG and ZBTB12 in regulation of LTR7/HERVH.

A: We found that the NANOG binding motif is significantly enriched in ZBTB12 binding loci identified by ChIP-seq (Supplementary Fig. 5d). Moreover, RNA-seq data after either ZBTB12 KD or NANOG KD showed the opposite phenotypes in LTR7/HERVH expression (Fig. 4i and j). Based on these results, we suggest that HEVRH expression is tightly regulated in a pluripotent state by both activation and repression mechanisms related to ZBTB12 and NANOG. However, we agree that detailed mechanisms for ZBTB12 and NANOG interaction are unclear. We rephrased the title of the paragraph and addressed this limitation in revised manuscript. Please see Page 13 Line 14 and Page 16 Line 20.

15. Page 16-eventually the LINC-ROR declines to control levels in the study in 6d, even though the state of the cells is quite different

A: As the reviewer pointed out, ZBTB12 KD delays LINC-ROR downregulation in the early stages of differentiation, while it declines to control levels at Day 10. These results suggest that ZBTB12 KD delays, but does not completely block LINC-ROR downregulation. Delayed LINC-ROR downregulation in the early stages is likely to influence pluripotency exit and differentiated cell fates.

16. Figure 6f-isolated fields showing a few cells are not convincing, the authors should carry out flow cytometry or some quantitative analysis. Same comment with Figure 6k.

A: The representative results of Fig. 5f and 5k (in the revised manuscript) are confirmed by image quantification and western blotting. Figure 5f is updated and the western blot images of 5k is provided as Supplementary Fig. 10n.

Reviewer #2 (Remarks to the Author):

In the manuscript entitled "ZBTB12-HERVH-LncRNA axis is a molecular barrier for dedifferentiation of human stem cells", Han et al claim that: 1. Transcription repressor ZBTB12 is required for the differentiation of human pluripotent stem cells (hPSCs) into three germ layers by acting as a molecular barrier for the dedifferentiation of hPSCs to more primitive naïve like state. 2. ZBTB12 repress the expression of HERVH, especially newly emerged and active ones. 3. ZBTB12 functions through HERVH overlapping lncRNAs including ESRG and LINC-ROR. These findings are interesting and novel, should be of interest to scientists in fields of stem cells, gene regulation and retrotransposon regulation and evolution. In particular, their findings may potentially lead to the forming of new concept on the domestication of ERVs. Their claims are generally supported by their data.

My major concerns are:

1. The claim on ZBTB12 acting as the barrier for the dedifferentiation of hPSCs needs to be tested in different differentiation conditions. For example, during embryoid body differentiation, neural differentiation or other directed differentiation conditions, do authors also see the upregulation of naïve markers upon ZBTB12 knockdown?

A: As suggested, we further assessed the expression of naïve marker genes in an embryoid body differentiation condition. Indeed, ZBTB12 KD showed significant upregulation of some naïve markers (NANOG, TFCP2L1, DNMT3L, DPPA5, HERVK), supporting our idea of stem cell dedifferentiation by ZBTB12 KD. The result is shown in Supplementary Fig. 4i.

2. The function of LINC-ROR and ESRG in the dedifferentiation process needs to be evaluated. Specifically, under spontaneous differentiation condition, whether overexpressing LINC-ROR or ESRG leads to the upregulation of naive state genes?

A: Under spontaneous differentiation condition, H9 hESCs overexpressing LINC-ROR showed some degree of upregulation of naïve state genes. However, the effect was not as strong as ZBTB12 KD, suggesting that other HERVH-associated lincRNAs may also contribute to the ZBTB12 KD phenotype. Please see Page 19 Line 18 and Supplementary Fig. 10h.

3. The authors claim that the function of ZBTB12 in blocking dedifferentiation is newly evolved in primate. This claim needs further evidence. What happens if knocking Zbtb12 in mouse epiblast stem cells? Answering this question may also help to clarify the function of human specific lincRNAs in dedifferentiation.

A: To test the effect of Zbtb12 on retrotransposon expression in mouse epiblast stem cells (mEpiSCs), we conducted bulk RNA sequencing after Zbtb12 KD. In stark contrast to the results in hESCs, Zbtb12 KD did not alter global transcription of mouse retrotransposons including LINE, SINE, and LTR (Supplementary Fig. 6d, e), supporting our hypothesis that ZBTB12-mediated HERVH suppression has emerged during primate evolution. Consistently, Zbtb12 KD did not show dramatic effects on mEpiSC differentiation (Supplementary Fig. 6h-j). Please see Page 16 Line 1.

My minor concerns include:

1. The title may need to be rephrased if further evidence is not provided. First, the role of HERVH and lincRNA in the dedifferentiation is not defined; second, human stem cells should human pluripotent stem cells.

A: As suggested, we rephrased the title.

2. Page 11, "Collectively, these results suggest that ZBTB12 serves as a molecular barrier for a naïve-like pluripotent state and this role of ZBTB12 is essential for successful differentiation of hESCs."; "is essential" better to be "may be".

A: Edited as suggested.

3. The cooperative model of ZBTB12 and NANOG in regulating HERVH expression needs to be further clarified. Do ZBTB12 and NANOG antagonize each other in binding HERVH sites? Or do they bind together?

A: We found that the NANOG binding motif is significantly enriched in ZBTB12 binding loci identified by ChIP-seq (Supplementary Fig. 5d). Moreover, RNA-seq data after either ZBTB12 KD or NANOG KD showed the opposite phenotypes in LTR7/HERVH expression (Fig. 4i and j). Based on these results, we suggest that HEVRH expression is tightly regulated in a pluripotent state by activation and repression mechanisms. However, we agree that detailed mechanisms for ZBTB12 and NANOG interaction are unclear. We rephrased the title of the paragraph and addressed this limitation in revised manuscript. Please see Page 13 Line 14 and Page 16 Line 20.

4. Figure 4f is not consistent with Figure 4h. It could be that authors used different normalization methods. By the way, how data are normalized should be described in figure legends. For example, Data were normalized to Day 0 shControl.

A: The data are normalized to Day0. We added this information in the figure legend.

5. Figure 5a, a scale for the peak file will help to judge how far ZBTB12 binding site is away from LINC-ROR TSS.

A: Scale is presented as suggested. The figure is moved to Supplementary Fig 8a.

6. Figure 5f, pLV_VPA samples should be shown. If the expression of LTR7 et al is significantly upregulated by VPA, the claim on ZBTB12 functioning through recruiting HDAC is flawed. If ZBTB12 works through HDAC pathway, after knocking down ZBTB12 in hESCs, VPA treatment should have minimum impact on the expression of LTR7, unless there exists other mechanism to recruit HDAC to LTR7 locus.

A: The effect of VPA in control (pLV) was tested as suggested. In accord with reviewer's speculation, the effect was insignificant or mild. The figure is updated and shown as Supplementary Fig. 9e.

7. The crosstalk between ZBTB12 and SIN3A/HDAC complex needs to be further clarified. ChIP using antibodies to SIN3A/HDAC in ZBTB12 knocking down cells may help to build a model that ZBTB12 recruits SIN3A/HDAC to repress HERVH.

A: As reviewers suggested, we conducted ChIP-seq experiments using 1) anti-H3K27ac and 2) anti-HDAC1 antibodies after ZBTB12 KD (Supplementary Fig. 9 and Supplementary Table 12). First, we assessed the level of H3K27ac, one of the major substrates of SIN3A/HDAC complex, nearby ZBTB12 binding peaks (from our ZBTB12 ChIP-seq data). As shown in Supplementary Fig. 9d, H3K27ac level is significantly increased after ZBTB12 KD, supporting our hypothesis (Pearson's Chi-squared test, X-squared = 4.4728, df = 1, p-value = 0.03444). However,

HDAC1 ChIP-seq showed no significant difference between control and ZBTB12 KD condition nearby ZBTB12 binding peaks in H9 hESCs (Pearson's Chi-squared test, X-squared = 0.57376, df = 1, p-value = 0.4488).

This may be explained by previous reports. First, transcriptionally active HERVHs are reported to demarcate topologically associating domains (TADs) in hPSCs (2019, Zhang et al., Nature Genetics), where dynamic epigenetic modifications including HDAC1-driven deacetylation of H3K27ac occur (2020, Qiao et al., Molecular Therapy). Our original data (Fig. 4) and new ChIP-seq of H3K27ac (Supplementary Fig. 9d) indicate increased transcriptional activity of HERVHs after ZBTB12 knockdown. This may result in global change in TADs and the binding of epigenetic modulators around those TADs. Secondly, the SIN3A/HDAC corepressor complex has been shown to directly bind with NANOG to promote pluripotency (2017, Saunders et al., Cell Reports). Given that our ZBTB12 ChIP-seq loci is highly enriched with NANOG (Supplementary figure 5d), it is likely that HDAC1 recruitment to the ZBTB12 binding loci is dynamically regulated by multimodal mechanisms such as negative feedback loop. Thus, our HDAC1 ChIP-seq result that did not show significant loss of binding activity after ZBTB12 KD does not necessarily mean that HDAC1 is not recruited by ZBTB12. However, we acknowledge that our argument is not strong enough with the current data to conclude the mechanism of ZBTB12-driven HERVH transcriptional repression. Therefore, we provide the entire data as Supplementary Figures 8 and 9 in the revised manuscript.

8. For experiments in Figure 1e, 1h, 6g, 6k, it would be nice to have more quantitative data such as qPCR or Western.

A: As suggested, we conducted western blotting to confirm the original results. The figures are updated: for Fig. 1e → Fig. 1f; for Fig. 1h → Supplementary Fig. 2t, for Fig. 5g (Fig. 6g in the original manuscript) → Supplementary Fig. 10g; for Fig. 5k (Fig. 6k in the original manuscript) → Supplementary Fig. 10n.

9. Supplementary Figure 7h, the data for shA and shB are not consistent, see the expression of NANOG in shControl.

A: We have been experiencing variation in differentiation speed across independent differentiation experiments, which could be at least partially attributed to different lots of reagents such KnockOut Serum Replacement (KOSR). We repeated the experiment and obtained consistent NANOG downregulation in shControl upon differentiation. The figure is updated in Supplementary Figure 10l.

10. Is ZBTB12 downregulated in naive hPSCs and upregulated during differentiation?

A: We analyzed ZBTB12 protein expression by immunostaining in naïve-like and primed hPSCs, and differentiated cells. Interestingly, ZBTB12 showed the highest expression in naïve-like hPSCs. As hPSCs proceeded from a naïve through a primed to a differentiated state, ZBTB12 expression gradually decreased (Supplementary Fig. 5o), which resembles the HERVH expression (Supplementary Fig. 5m). These results are consistent with our finding that ZBTB12 regulates transcriptionally active HERVH loci (Supplementary Fig. 8) and suggest that ZBTB12 contributes primarily to fine-tuning the expression of active HERVH loci. Cell state-dependent HERVH expression is likely to be regulated by core pluripotency genes such as NANOG. Overall, these results accord with our conclusions about ZBTB12 functioning as a barrier, but not a cell fate driver. Please see Page 15 Line 14, and Supplementary Fig. 5o.

11. Data point from same batch of experiment may be labelled by the same color or same shape.

A: Updated as suggested. Please see Fig. 1d, Fig. 5e, Supplementary Fig. 2f, Supplementary Fig. 3n and Supplementary Fig. 10f.

12. Statistic test should not be made for n = 2 (e.g. Supplementary Figure 7d).

A: Experiments were now triplicated and updated.

13. For multiple comparison, appropriate statistic tests should be applied. Current version used Student's t-test throughout the manuscript.

A: In the revised manuscript, ANOVA is used for multiple comparisons.

Reviewer #3 (Remarks to the Author):

Han et al are going after a highly interesting subject: regulation of human pluripotency. Their pipeline reveals ZBTB12 as a novel candidate that controls the exit from a naive/naive-like pluripotency, thus removes a barrier from cell differentiation. Their single cell RNA sequencing reveals that ZBTB12 was essential for three germ layer differentiation by blocking dedifferentiation of hPSCs toward a more primitive state. Mechanistically, they suggest that ZBTB12 serves as a master repressor of human endogenous retrovirus H (HERVH), previously implicated in regulating pluripotency in primates. The authors claim at identifying the ZBTB12-HERVH-lncRNA axis as a molecular machinery that safeguards the unidirectional transition of metastable stem cell fates toward developmentally

advanced states. The story is highly interesting, but only partially supported.

To identify novel regulators of pluripotency, the authors use a strategy that is not dependent on expression changes of TFs, and could be used to precisely map transcription start sites (TSSs). This approach identifies ZBTB12. ZBTB12 is a phylogenetically conserved protein (here, shown between mice and human), and has a broad expression in multiple cell types and tissues. Notably, its role in regulating pluripotency is not known.

The first part of the manuscript is relatively solid. The authors convincingly demonstrate that ZBTB12 is a novel gene that controls exit from naivety.

Importantly, phylogenetically, HERVK is much younger than HERVH. HERVH and HERVK cannot be mentioned together, because they are not regulated the same way and they are marking different cell types. Notably, the study seems to reveal a different type of naivety upon KD ZBTB2, analyzed by single cell RNA-seq, or when they use the LIF, 3i protocol to induce a naive state. In the second case, HERVK is upregulated much stronger than TFCP2L1. What about the HERVH-related products in this condition? Are these different types of cells?

A: Compared to HERVK, HERVH showed mild upregulation in the 3iL condition. However, ZBTB12 KD significantly increased the expression of both HERVH and HERVK in the 3iL condition (Fig. 3i). To further define the cell type of ZBTB12 KD cells, we performed additional scRNA-seq analysis with published scRNA-seq data from human embryos and naïve and primed cells. The dedifferentiated cells by ZBTB12 KD (Day 8 shZBTB12) clustered together with naïve hPSCs (see Fig. 3g, h). However, we agree the potential difference between 3iL-induced and ZBTB12 KD-induced naïve-like cells and we clarified this issue in the revised manuscript. Please see Page 13 Line 1.

Recent studies suggest that HERVK is upregulated in artificially induced naivety (where HERVH is not highly expressed), while high HERVH expression marks a subpopulation of naive-like cells, naturally present in PSC cultures. The relationship between HERVK vs HERVH should be clarified in the manuscript.

A: We clarified this issue in the revised manuscript. Please see Page 13 Line 1.

The second part, where the authors try to decipher the underlying mechanism, is far less convincing. They claim that "As a downstream mechanism, ZBTB12 binds to and transcriptionally suppresses HERVH." However, this is not well supported by their data.

A: We acknowledge the limitation and we have revised the title and softened our claims by limiting this report to the data we have without drawing broader conclusions.

The study supports the previously suggested role of HERVH and HERVH-derived transcripts/lncRNA in regulating naive-like pluripotency. The control of the HERVH-mediated pluripotency regulation is not fully understood. While, the set of TFs, whose expression upregulates HERVH transcription in pluripotent stem cells is reported, the suppression of the regulatory network is yet to be deciphered. Because ZBTB12 is expressed antagonistically with HERVH, a HERVH-derived gene ESRG and lncROR, the authors focused on this relationship. They present evidence that upon depletion of ZBTB12 RNA, pluripotency is affected. However, whether this effect can be linked to a direct control over HERVH by ZBTB12, is not supported well.

A: We showed that ZBTB12 suppresses the expression of HERVH and HERVH-overlapping lncRNAs (Fig 4g, Supplementary Fig. 10b). In particular, ZBTB12 binds to the promoter region of LINC-ROR, suggesting direct transcriptional regulation (Supplementary Fig. 10a). Furthermore, delayed pluripotency exit by ZBTB12 depletion was rescued by knocking down this lncRNA (Fig 5k, Supplementary Fig. 10n). These results suggest that the effect of ZBTB12 KD is at least partially mediated by upregulation of the HERVH-overlapping lncRNA. However, we agree that it is not clear whether the effect of ZBTB12 KD on pluripotency exit is due to a direct control over HERVH. Therefore we have revised our claims accordingly.

"Significant correlation of HERVH expression with the overlapping lncRNAs was observed after ZBTB12 KD (Pearson correlation analysis, $R=0.53$, $P<0.001$), suggesting that ZBTB12 regulates expression of the lncRNAs through LTR7/HERVH (Fig. 6a, Supplementary Table 6). - These data present correlation, but not a direct evidence.

A: We agree on this comment and has edited the sentence in the revised manuscript (Page 18 Line 22).

Although ZBTB12 binds DNA (ChIP-seq), this binding does not seem to be specific to HERVH sequences (>200 copies upregulated in pluripotency). In fact, the binding of ZBTB12 on HERVH elements is not significant: The authors find that "...ZBTB12 binding sites were frequently found in up- or downstream of HERVH loci." ; "... among 70 ZBTB12 ChIP-seq peaks associated to full length HERVH loci (Supplementary Table 7), only three peaks were located on LTR7."

Consider, if we randomly distribute ChIP-seq peaks on human genome, it is very likely that ~10-20 HERVH loci would be occupied by chance. Thus, it is not statistically significant that ZBTB12 binds and regulates HERVH-derived sequences and pluripotency.

We agree that unlike KRAB-ZFPs, ZBTB12 does not seem to directly recognize HERVH sequences. KRAB-ZFPs have been co-evolved with LTR retrotransposons suggesting an evolutionary arms race. In this setting, the evolution of KRAB-ZFPs has been driven to direct recognition of LTR sequences to efficiently suppress the newly invaded genetic elements. However, ZBTB12 is evolutionarily conserved through vertebrates, while its target, HERVH, extensively invaded human genome during primate evolution. This unique relationship between ZBTB12 and HERVH explains the observation that ZBTB12 binding sites are frequently found in up- or downstream of HERVH loci. ZBTB12 binding sites nearby HERVH could be newly evolved to restrict HERVH expression or HERVH insertions nearby pre-existing ZBTB12 binding sites have been selected during evolution. Although we identified only 70 ZBTB12 binding sites nearby full-length HERVH loci (within 10 kb), ZBTB12 KD in hESCs showed global upregulation of almost all HERVH loci, which is probably due to both direct and indirect effects of ZBTB12 on HERVH regulation. Therefore, we clarified in the revised manuscript that direct regulation by ZBTB12 is restricted on specific HERVH loci. Please see Page 2 Line 12 and Page 5 Line 5.

Let's take the possibility that ZBTB12 is specifically binding on three copies of HERVH. Even the identified 3 peaks map to different parts of HERVH sequences, arguing against phylogenetic conservation.

A: Please see our response to the above comment. We agree that ZBTB12 does not specifically recognize HERVH sequences. This point is further discussed in Page 24 Line 13.

Is it possible that ZBTB12 controls only these copies, and these are the key regulators? Notably, the study is not following up on these particular copies, but on ESRG and LINC-ROR. Indeed, the HERVH products (ESRG, LINC-ROR) were already shown by others to effect pluripotency.

A: Among HERVH functions, we focused on lncRNAs in this study. Our criteria of selecting potential lncRNAs that mediate pluripotency regulation were as follows. In all identified lncRNAs overlapped with LTR7/HERVH loci in human genome, candidate lncRNAs should be detected in our RNA-seq data (56 out of 260 lncRNAs were detected). And we selected the lncRNAs that are either the most highly expressed (based on CPM in RNA-seq) or the most highly upregulated by ZBTB12 KD, which are ESRG (1584.368 CPM in Scramble sample, average CPM of the 56 lncRNAs is 59.88) and LINC-ROR (5.54-fold upregulation by ZBTB12 KD, average fold change of the 56 lncRNAs is 1.86), respectively. Although there are two lncRNAs (AL365259.1 and LINC02523) associated with the LTR7 elements where ZBTB12 ChIP-seq peaks are directly located, these lncRNAs failed to pass our criteria. CPM values for these two lncRNAs were 1.82 and 2.25 respectively with low Log2FC (1.27 and 0.55 respectively).

Overall, it is not clear what might provide the specificity of the putative ZBTB12-mediated HERVH (if any) regulation. Their hypothesis that ZBTB12 controls HERVH via recruiting SIN3A/HDAC1 complex is interesting, but not convincing either. The study demonstrates an immunoprecipitation of ZBTB12-SIN3A and ZBTB12-HDAC1. This might suggest that similar to ZBTB16 (PLZF), ZBTB12 is involved in a protein-protein interaction with the SIN3A/HDAC1 complex. The hypothesis that ZBTB12 works in conjunction with the SIN3A/HDAC complex is likely, but again the data does not support direct connection with HERVH regulation. A dissection of a complex, bound to a HERVH locus could identify the components. This approach might substantiate that SIN3A/HDAC is recruited to a specific HERVH locus by ZBTB12.

A: In the revised manuscript, we performed ChIP-seq with anti-HDAC1 and anti-H3K27ac antibodies in H9 hESCs (Supplementary Fig. 9 and Supplementary Table 12). A k-mean clustering analysis revealed that 68.7% of ZBTB12 binding sites nearby full-length HERVH (within 10 kb) are present in Cluster 4,5,6, and 7, where ZBTB12 binding sites are co-occupied by HDAC1 (Supplementary Fig. 9a). These results together with the co-IP data between ZBTB12 and HDAC1 proteins (Supplementary Fig. 8e and 8g) suggest that HDAC1 is at least partially involved in ZBTB12-mediated HERVH repression. However, ZBTB12 KD did not affect HDAC1 binding to ZBTB12 binding sites (Supplementary Fig. 9d), which can be in part due to complex feedback regulation of epigenetic factors. We acknowledge that our argument is not strong enough with the current data to conclude the mechanism of ZBTB12-driven HERVH transcriptional repression. Therefore, we provide the entire data as Supplementary Figures 8 and 9 in the revised manuscript but do not draw the conclusion anything concerning the mechanism of ZBTB12-driven regulation of HERVH transcriptional repression.

The evolutionary aspect might be interesting. The study shows that the mouse ZBTB12 is able to inhibit HERVH expression, however it might be interesting to find out whether ZBTB12 also controls the exit from naivety in mice. Is the proposed regulatory function of ZBTB12 specific to human/primates?

A: As suggested, we performed bulk RNA-seq and differentiation assays in mouse epiblast stem cells (mEpiSCs), which are a mouse counterpart of hESCs. Unlike hESCs, Zbtb12 KD in mEpiSCs showed no effect on the expression of retrotransposons such as LINE, SINE, and LTR (Supplementary Fig. 6d, e). Furthermore, mEpiSC differentiation was not significantly affected by ZBTB12 KD (Supplementary Fig. 6h-j). These results suggest that ZBTB12-mediated control over HERVH and pluripotency exit is a human/primate-specific feature.

The sequence of the ZBTB12 is conserved between mouse and human. Are the aa changes under selection? Does ZBTB12 binds differentially to genomic DNA in mice?

A: The amino acid sequences of ZBTB12 are highly conserved between mouse and human with a few amino acid changes that are localized outside of major functional domains such as Zinc finger and BTB domains. These results suggest that mouse ZBTB12 probably recognizes the same DNA motif of human ZBTB12. The evolutionary conservation was further confirmed using KaKs_Calculator2.0 as shown below (Wang et al., (2010). KaKs_Calculator 2.0: a toolkit incorporating gamma-series methods and sliding window strategies. Genomics Proteomics Bioinformatics, 8(1), 77-80). Please see Page 15 Line 19.

Software: KaKs_Calculator2.0
Result: Ka/Ks=0.0174919, P-Value (Fisher)=3.63453e-90
Conclusion: Ka/Ks<1, negative purifying selection

As we mentioned above, Zbtb12 KD in mEpiSCs did not show any significant effects on retrotransposons and stem cell differentiation, suggesting that Zbtb12 has acquired the role of LTR regulation as HERVH invaded human genome during primate evolution.

Further issues:

"... mouse PSCs cultured with serum exhibit substantial intrapopulation heterogeneity with a mixture of naïve- and primed-like cells, which represents a dynamic equilibrium of cells transitioning between distinct metastable states^{9,10}."

- Importantly, human PSC cultures are also heterogenous, however they have much lower percentage of naive-like cells.

A: We added this point in the revised manuscript. Please see Page 3 Line 10.

"ZBTB12-mediated suppression drives efficient shutdown of young and transcriptionally active HERVH loci upon pluripotency exit. These results suggest that ZBTB12 is engaged in regulating young and transcriptionally active HERVH loci."

- Nota that young is not equal to transcriptionally active! The HERVH cannot be considered as a young element. Perhaps, the authors mean less diverged?

A: We edited "young HERVHs" to "less-diverged HERVHs". As you can see in Supplementary Fig. 5j, k, we revealed that the highly expressed HERVHs (transcriptionally active, top 200 loci) exhibited lower divergence time than the ones with low expression ($p=2.2e-16$).

Figure 3g.

"Treating hESCs with LIF and 3i was sufficient to transiently activate the expression of naïve pluripotency-related genes and ZBTB12 KD further promoted naïve gene activation (Fig. 3g)."

- Show ESRG and LINC-ROR on the figure. Curiously, HERVK is much more induced than TFCEP2L1 or NANOG..!

A: The expression levels of LINC-ROR and ESRG were measured as suggested (Fig. 3i). LINC-ROR was increased by naïve state induction and this process was significantly facilitated by ZBTB12 KD. However, ESRG showed mild to no difference by ZBTB12 KD. Values were normalized to day 0.

In our transient induction of a naïve-like state by 3iL, the expression of TFCEP2L1 and NANOG is not significantly upregulated until Day 2 compared to other naïve markers such as HERVK. It is probably due to the short induction period that we used in this study.

"Despite the high expression level in hESCs, ESRG KD did not have any significant effects on the expression of core pluripotency genes (Supplementary Fig. 7f, g)."

- Confusing! The cited figures show significance.

A: The asterisks in Supplementary Fig. 7f in the original manuscript indicate the significance of ESRG knockdown level after shRNA transduction. Immunostaining of OCT4 and NANOG in Supplementary Fig. 7g (in the original manuscript) shows no difference after ESRG KD. Please note that these figures are now Supplementary Fig. 10j, k in the revised manuscript.

"Focusing on LTR7/HERVH, we identified 280 lncRNAs overlapped with LTR7/HERVH loci in the human genome (Supplementary Table 8). Among them, 56 lncRNAs were detected in our RNA-seq data with counts per million (CPM) higher than one (Supplementary Table 9)". - Similar data has been already reported in 1.

A: We added this point in the revised manuscript (Page 18 Line 20).

Reviewer #4 (Remarks to the Author):

The authors identify ZBTB12 as pluripotent stem cell associated transcription factor. They find that ZBTB12 knock-down impairs exit from pluripotency and ZBTB12 overexpression allows better differentiation. Using ChIP-seq of a

tagged ZBTB12 protein, they identify genome-wide binding sites. Interestingly, numerous binding sites appear associated with HERV sequences. ZBTB12 knock-down results in higher HERV expression (with focus on LTR7 driven HERVs), suggesting a repressive role of ZBTB12. Interestingly, there is an inverse correlation between Nanog knock-down and ZBTB12 knock-down on HERV expression, suggesting that ZBTB12 counteracts Nanog activation of HERVs. The authors find a partial overlap between ZBTB12 peaks and HDAC/SIN3A binding sites, suggesting a mechanisms for transcriptional repression. Furthermore ZBTB12 interacts with HDAC/SIN3A. Finally, the authors study HERV-associated lncRNAs and their role in pluripotency exit. They show that two lncRNAs, LINC-ROR and ESRG influence pluripotency exit. The authors propose that ZBTB12 represses HERV sequences and HERV associated lncRNAs. Loss of ZBTB12 results in HERV and lncRNA derepression, which somehow impairs pluripotency exit.

In its current state, the manuscript suggests an important role of ZBTB12 in human pluripotent stem cells. However, the mechanism is very loosely established. The role of ZBTB12 in HERV repression is not fully clear. There are ZBTB12 peaks in some distance to LTR7/HERV sequences, however, how this binding affects HERV transcription at a distance is unclear. Also, the transcriptional regulation of HERV-associated lncRNAs is not well described. Are these lncRNAs actually all driven from LTR7 promoters, as suggested in the screenshot for LINC-ROR? This would rather indicate that HERVs have evolved into regulatory sequences of these lncRNAs and the regulation by ZBTB12 is not so much towards HERVs, but rather towards specific transcripts which by chance utilize LTR promoters. In this sense the title suggesting a ZBTB12-HERVH-lnc-RNA axis is a bit misleading.

A: We agree that unlike KRAB-ZFPs, ZBTB12 has not evolved to directly regulate HERVs. KRAB-ZFPs have co-evolved with LTR retrotransposons, which is well documented as an evolutionary arms race. In stark contrast, ZBTB12 is evolutionarily conserved through vertebrates, while its target, HERVH, extensively invaded human genome during primate evolution. As the reviewer mentioned, it is likely that new HERVH insertions that by chance happened nearby pre-existing ZBTB12 binding sites have been selected during evolution. This idea provides some explanation to our observation that ZBTB12 binding sites are frequently found in up- or downstream of HERVH loci. We clarified this issue in the revised manuscript (Page 2 Line 12 and Page 5 Line 5) Although our ZBTB12 ChIP-seq identified 70 peaks nearby full-length HERVH loci, RNA-seq after ZBTB12 KD showed upregulation of most HERVH loci in hESCs without major alteration in protein coding genes. These results suggest that the global increase of HERVH by ZBTB12 KD can be attributed to both direct and indirect mechanisms. Therefore, we agree on the reviewer's comment on the title. Please see our new title in the revised manuscript.

Based on the authors demonstrating the effect of two lncRNAs on pluripotency, the potential mechanism of action of these lncRNAs should be discussed more extensively.

A: The suggested points were discussed in the revised manuscript. Please see Page 23 Line 5.

Specific comments:

(1) In order to connect ZBTB12 with transcriptional regulation a clear intersection between ZBTB12 binding sites and transcripts in the vicinity which change expression in ZBTB12 knock-down should be established.

A: We assessed the effects of ZBTB12 KD on LTR7/HERVH loci found within 10 kb of ZBTB12 binding sites (LTR7_ChIP) as well as protein coding genes (PCGs) whose TSSs are located within 1 or 2 kb of ZBTB12 binding sites (PCGs_1k or PCGs_2k). ZBTB12 KD significantly upregulated the expression of LTR7_ChIP, but not of PCGs. Currently, we do not know how ZBTB12 specifically regulates LTRs and have addressed this limitation in the revised manuscript (Page 14 Line 21, Supplementary Fig. 5h, i). Given the global upregulation of HERVH loci after ZBTB12 KD, no significant difference between total LTR7 and LTR7_ChIP was observed after ZBTB12 KD. These results can be attributed to technical limitations regarding ChIP-seq in highly repetitive sequences and feedback mechanisms regulating HERVH transcription. For example, it is known that HERVH transcripts function as scaffolds for epigenetic modulators (e.g., p300) and active HERVH transcription contributes to creating boundaries of topologically associating domains (TADs) in human PSCs. Please see our discussion in Page 24 Line 20.

In the initial screen the authors identify ZBTB12 as potential transcription factor binding to TSS of the nanoCAGE transcripts. Could ZBTB12 binding to these sequences be confirmed in the ChIP-seq data?

A: In the nanoCAGE data, we identified 2335 TSS peaks specific to Day 0. Based on 300 bp flanking the TSS peaks, 153 out of 2335 TSS peaks were found to have the ZBTB12 binding motif. Among 153 TSS peaks with the ZBTB12 binding motif, 55 peaks (55/153; 35.9%) are located in LTR regions, suggesting that ZBTB12 could be a regulator of LTRs. When the 55 LTR peaks were compared to our ZBTB12 ChIP-seq peaks, we found 5 overlapping peaks (5/55; 9.1%) with 2kb window size. This limited overlap can be explained by technical limitations. First, LTR sequence redundancy across human genome make it challenging to fully detect ZBTB12 binding sites by ChIP-seq. Second, the nanoCAGE data include only 300 bp sequences flanking TSS peaks, which misses distal ZBTB12 binding sites. Nevertheless, the two independent techniques, nanoCAGE and ChIP-seq, suggest the regulatory role of ZBTB12 in LTR expression.

(2) The ZBTB12 ChIP-seq data need to be quality controlled better before they can be judged. The identified binding motif needs to be compared to the reference motif. The full result table of the de novo motif finding analysis needs to be provided. This also helps to judge the co-enriched motifs.

A: Among 3621 ChIP-seq peaks, 1083 peaks were found to have the ZBTB12 consensus sequence motif. The full result table of the de novo motif analysis is provided in the revised manuscript (Supplementary Table 7).

The binding sites (and genomic features on binding sites) should be compared to the published profile in HEK cells. A: Global genomic features between our ZBTB12-Flag ChIP-seq data in hESCs and the published ZBTB12-GFP ChIP-seq data in HEK cells are quite different as shown below, which is probably due to different cell types. However, both datasets consistently showed the enrichment of ZBTB12 binding sites in LTR (Fig. 4b and Supplementary Fig. 5c). These results further support the idea of ZBTB12 as an LTR regulator.

Genomic feature	Log2 Enrichment score in HEK cells	Log2 Enrichment score in hESCs
Exon	0.371	3.384
Intergenic	-0.074	-0.603
Intron	0.019	-0.254
Promoter-TSS	0.384	2.737
TTS	0.013	-0.02
LINE	0.211	-2.231
SINE	-0.15	-2.513
LTR	0.579	0.476

(3) Binding of ZBTB12 to HERV sequences needs to be more specifically investigated and presented. The example screenshot in Fig 4e demonstrates binding to an LTR7 promoter. However, only 7 peaks associate with LTR7 promoters, indicating that direct HERV binding is strongly under-represented. Later in the manuscript the authors come back to this point and argue that there are cis effects of distant ZBTB12 binding.

This relates to point (1) which genomic features are associated with ZBTB12 mediated transcriptional effects.

A: Please see our response to your comment (4).

Their final argument is that ZBTB12 regulates lncRNAs LINC-ROR and ESRG, so at least for these two examples, ZBTB12 binding and transcriptional effects should be shown in screenshots.

A: As suggested, screenshots for LINC-ROR and ESRG below are provided in the revised manuscript (Supplementary Fig. 10a, i).

(3) Transcriptional changes in HERVs should be better represented on the family level or on the level of individual transcripts (especially if there is a focus on specific lncRNAs) with baseMean vs log2fc.

A: In the original manuscript, we analyzed the expression of retrotransposon subfamilies (Fig. 4c) and LTR subfamilies (Fig. 4d) after ZBTB12 KD and observed the most dramatic upregulation of the LTR7/HERVH subfamily. We also provided a plot with baseMean vs log2fc. Please see Supplementary Fig. 5e.

Is there a transcriptional change on whole HERV families or is there effects on specific HERV integrations?
Fig4 C and D

A: The effect of ZBTB12 KD is specific to LTR7/HERVH subfamily because no significant transcriptional changes were observed in LINE, SINE, and other LTR subfamilies (Fig. 4c and d). Interestingly, ZBTB12 KD upregulated almost all LTR7/HERVH loci in hESCs (Fig. 4d), which is likely due to both direct and indirect effects of ZBTB12. As we mentioned above, epigenetic roles of HERVH transcripts and its transcription would underlie the indirect effects.

Which HERVs are bound by ZBTB12 (or indirectly affected in cis)?

A: Because ZBTB12 KD induced significant expression changes of LTR7/HERVH, but not other HERVs, we focused on LTR7/HERVH loci. Among full-length LTR7/HERVH loci, we identified 3 ZBTB12 ChIP-seq peaks that are directly localized on LTR7/HERVH loci and 67 peaks within 10 kb of LTR7/HERVH loci. Please see Supplementary Table 11.

Transcriptional changes in LTR7 in Fig 4f,g: which copies are detected with these primers?

A: LTR7, HERVH GAG and HERVH POL primers can detect 49, 423 and 320 HERVH loci, respectively. We provided the list of target genomic loci of each primer set as a Supplementary Table 8.

(4) Repression by ZBTB12 is suggested to be mediated by HDAC/SIN3A. Fig.5b showing co-occupancy between these factors and ZBTB12 clearly shows only partial overlap between these factors. To analyze these data more robustly, an overlap analysis (and k-means clustering) needs to be performed to identify ZBTB12 peaks which associate with HDAC and SIN3A (show example screenshots). These peaks should be intersected with ZBTB12 regulated transcripts. If there is a good overlap between ZBTB12 repressed transcripts and HDAC/SIN3A binding, it should be tested if recruitment of HDAC/SIN3A depends on ZBTB12 (and vice versa). Treatment of cells with HDAC inhibitors as done by the authors, is known to up-regulate HERVs and is not specific to ZBTB12 activity.

A: In the revised manuscript, we performed ChIP-seq with anti-HDAC1 and anti-H3K27ac antibodies in H9 hESCs (Supplementary Fig. 9 and Supplementary Table 12). A k-means clustering analysis revealed that 68.7% of ZBTB12 binding sites nearby full-length HERVH (within 10 kb) are present in C4, C5, C6, and C7, where ZBTB12 binding sites are co-occupied by HDAC1 (Supplementary Fig. 9). These results together with the co-IP data between ZBTB12 and HDAC1 proteins (Supplementary Fig. 8e-g) suggest that HDAC1 is at least partially involved in ZBTB12-mediated HERVH repression. Although ZBTB12 KD significantly induced H3K27ac levels in ZBTB12 binding sites, HDAC1 binding was not significantly altered (Supplementary Fig. 9d). These results can be in part due to complex feedback regulation of epigenetic factors, as discussed in Page 24 Line 20. We acknowledge that our argument is not strong enough with the current data to conclude the mechanism of ZBTB12-driven HERVH transcriptional repression. Therefore, we provide the entire data as Supplementary Fig. 8-9 in the revised manuscript.

(5) most figures are hard to read, please increase font size in all figures

A: Figures are resized as suggested.

Reviewer #1 (Remarks to the Author):

The authors have done a thorough job addressing the points raised in my review, and the manuscript has been strengthened considerably. Ultimately more work will be required to elucidate the role of ZBTB12/HERVH in pluripotency state transitions, but the present manuscript has opened up new avenues of investigation into this important question.

Reviewer #2 (Remarks to the Author):

The authors have addressed all my questions. I support publishing the manuscript in Nature Communications.

Reviewer #4 (Remarks to the Author):

The authors have appropriately addressed several of my concerns. However, there are still unresolved issues and based on their new analyses I have additional concerns that need to be addressed. The two major concerns are the quality/robustness of the ZBTB12 ChIP-seq peaks and the identification of ZBTB12 targets.

(1) The authors show that ZBTB12 is an important factor in hESCs. Knockdown of ZBTB12 results in impaired hESC differentiation (or pluripotency exit). What is still largely unclear is the role of ZBTB12 in this context. The authors addressed this by ChIP-seq analysis. However, the ZBTB12 ChIP-seq profile appears not very robust. Their new Motif analysis showed that only ~30% of ZBTB12 peaks contain the ZBTB12 motif. The de novo analysis was only partially shown with the summary table missing, so it could not be judged if an alternative motif would be more strongly enriched. For transcription factors, ~30% motif in peaks is very low, suggesting that the majority of peaks might be noise. This can also be seen in the screenshots of Figure S10i, where ZBTB12 peaks in the ESRG locus are close to noise levels (compared with the peak in S10a (LINC-ROR locus). Based on the questionable peak detection it is critical to generate replicate ChIP-seq datasets for ZBTB12 and to define high confidence peaks based on the intersection of datasets. Analyses using ZBTB12 peaks need to be repeated with the high confidence peaks.

(2) What are the directly regulated ZBTB12 targets in hESCs? Again, this analysis critically depends on having high confidence ZBTB12 binding sites that can be intersected with genes and HERVs in the vicinity. Here it is important to define the same target interval, for example ZBTB12 peaks in 10kb vicinity to gene TSS or HERV LTRs. Then it would be good to systematically evaluate the transcriptional effects in the context of ZBTB12 high confidence peaks. (a) how many genes are regulated (up/down) and how many of these have ZBTB12 binding sites in vicinity. (b) how many HERV families are regulated (up/down) and how many of these families have ZBTB12 enrichment (include multimapping reads to assess whole families)? (c) how many single HERV integrations are regulated (exclude multimapping reads and restrict to mappable elements) and how many have ZBTB12 peaks in vicinity? (d) If LTR7 elements are major targets, please show some examples in genome browser screenshots. The example in Fig. 4e needs to be expanded to better identify the transcription unit; also the ZBTB12 peaks appear very close to background based on the scale (in comparison with Fig S10a).

This analysis will help to better judge if ZBTB12 could regulate genes or HERVs or both. In particular HERV regulation appears difficult to follow based on the current analysis. In Figure S5e, the basemean vs log₂fc plot for single HERV loci (I assume) show a clear shift towards upregulation, suggesting that almost all HERVs are regulated. This would not correspond with the notion of the authors that only LTR7 HERVs are regulated. There might be a normalization problem here, so it would be important to include elements which are not regulated and which should form a cloud around log₂fc of 0. Another possibility is that most of the regulation (HERVs/genes) is explained by the cells assuming a naive-like state upon ZBTB12 KD. In this scenario, ZBTB12 KD triggers this

transition, but majority of regulation might be indirect due to the changed transcriptional network.

Additional comments:

The quality of the NANOG ChIP-seq profile is also questionable, at least the signals shown in Figure S10a/i look very noisy. It is always important to indicate identified peak positions for TFs in the genome browser views. Also it is important to demonstrate pos/neg control regions to judge background/noise. For claiming that there is antagonism between NANOG and ZBTB12 in HERV regulation, HERV target identification of NANOG needs to be more robust.

The RNA-seq tracks in FigS10a/i also look suspicious. There is very low correlation between the signals and the gene/LincRNA annotation. As these transcripts were used in functional assays it would be important to clarify the exact identity of the regulated transcript (spliced/unspliced/internal promoter used in LINC-ROR?).

Reviewer #5 (Remarks to the Author):

For clarity to the authors, I have come into this review process mainly to assess the rebuttal of the points raised by reviewer #3. I have read the manuscript with interest and congratulate the authors on an extensive and well conducted study. I see no major technical issues in this revised version, and can appreciate from the rebuttal that conclusions have been toned down to better reflect what the data show.

The authors have gone through substantial lengths to address the comments from reviewer #3, and the changes made to the manuscript look to be appropriate to me. I only have a relatively minor comment regarding the point on the non-random binding of ZBTB12 to HERVH loci. The reviewer had suggested a comparison to a random control. This seems to have been done for LTRs (Fig. 4b), although methodological details are missing, and these should be added in my opinion. A similar analysis could be done for HERVH specifically, to ask if the observed ZBTB12 peak distribution (overlapped and within 10kb of HERVH) is indeed non-random.

Miguel Branco

REVIEWER COMMENTS

Reviewer #1 (Remarks to the Author):

The authors have done a thorough job addressing the points raised in my review, and the manuscript has been strengthened considerably. Ultimately more work will be required to elucidate the role of ZBTB12/HERVH in pluripotency state transitions, but the present manuscript has opened up new avenues of investigation into this important question.

> We appreciate this reviewer for considering our work as a new avenue into a crucial biological question.

Reviewer #2 (Remarks to the Author):

The authors have addressed all my questions. I support publish the manuscript in Nature Communications.

> We appreciate this reviewer for supporting the publication of our work in Nature Communications.

Reviewer #4 (Remarks to the Author):

The authors have appropriately addressed several of my concerns. However, there are still unresolved issues and based on their new analyses I have additional concerns that need to be addressed. The two major concerns are the quality/robustness of the ZBTB12 ChIP-seq peaks and the identification of ZBTB12 targets.

> With new bioinformatics analyses, we have addressed the additional concerns raised by Reviewer #4. Please see below for our responses to the specific comments.

(1) The authors show that ZBTB12 is an important factor in hESCs. Knockdown of ZBTB12 results in impaired hESC differentiation (or pluripotency exit). What is still largely unclear is the role of ZBTB12 in this context. The authors addressed this by ChIP-seq analysis. However, the ZBTB12 ChIP-seq profile appears not very robust. Their new Motif analysis showed that only ~30% of ZBTB12 peaks contain the ZBTB12 motif. The de novo analysis was only partially shown with the summary table missing, so it could not be judged if an alternative motif would be more strongly enriched. For transcription factors, ~30% motif in peaks is very low, suggesting that the majority of peaks might be noise. This can also be seen in the screenshots of Figure S10i, where ZBTB12 peaks in the ESGR locus are close to noise levels (compared with the peak in S10a (LINC-ROR locus). Based on the

questionable peak detection is critical to generate replicate ChIPseq datasets for ZBTB12 and to define high confidence peaks based on the intersection of datasets. Analyses using ZBTB12 peaks need to be repeated with the high confidence peaks.

> To clarify whether it is a technical issue concerning our ZBTB12 ChIP-seq, we analyzed the ZBTB12 ChIP-seq data (GSE58341) done in HEK293 cells by an independent research group. Consistent with our results, 33.05% of the peaks contained the ZBTB12 motif (NGNTCTAGAACCNGV) (Please see updated Supplementary Table 7), suggesting that it could be a biological feature of ZBTB12. When we considered only the 8-bp core motif (CTAGAACC), 71.36% of our ZBTB12 ChIP-seq peaks contained the motif. Furthermore, analysis of our ZBTB12 ChIP-seq peaks by known transcription factor motifs revealed ZBTB12 with the most significant p-value. These results demonstrate the reliability of our ZBTB12 ChIP-seq results.

> As the reviewer pointed out, the ZBTB12 peaks in the ESRG locus did not pass our criteria for peak calling, suggesting that ESRG is an indirect target of ZBTB12. We clarified this in the revised manuscript (Page 20 Line 5).

> To validate our results with high-confident peaks, we re-analyzed the enrichment analysis only using the ZBTB12 peaks with motif and confirmed that the ZBTB12 peaks are specifically enriched in LTR retrotransposons among repetitive genomic regions (Supplementary Fig. 5c).

(2) What are the directly regulated ZBTB12 targets in hESCs? Again, this analysis critically depends on having high confidence ZBTB12 binding sites that can be intersected with genes and HERVs in the vicinity. Here it is important to define the same target interval, for example ZBTB12 peaks in 10kb vicinity to gene TSS or HERV LTRs. Then it would be good to systematically evaluate the transcriptional effects in the context of ZBTB12 high confidence peaks.

> As suggested, we used the same target interval (10 kb) and ZBTB12 high-confident peaks (with motif) to compare the transcriptional effects of ZBTB12 knockdown on LTR7/HERVH and protein-coding genes. Consistent with our original results, LTR7/HERVH loci, but not protein-coding genes, within 10 kb of the ZBTB12 high-confident peaks were transcriptionally upregulated by ZBTB12 KD (Supplementary Fig 5k,l).

(a) how many genes are regulated (up/down) and how many of these have ZBTB12 binding sites in vicinity.

> 152 DEGs (120 up- and 32 down-regulated) were found after ZBTB12 knockdown. Among those, 12 genes were found in the 10kb vicinity of ZBTB12 ChIP-seq peaks (10 up and 2 down). When only ZBTB12 high-confident peaks (with motif) were analyzed, 7 DEGs were found in the 10kb vicinity (6

up and 1 down). Please see the DEGs table for the detail (Supplementary Table 9).

(b) how many HERV families are regulated (up/down) and how many of these families have ZBTB12 enrichment (include multimapping reads to assess whole families)?

> As suggested, we analyzed the expression of HERV families, including multi-mapping reads. Among all LTR subfamilies, ERV1, which includes LTR7/HERVH, showed the most significant upregulation and enrichment score after ZBTB12 knockdown (Fig. S5g).

LTR subfamilies	Enrichment Score
ERVK	0.86
ERVL	0.63
ERVL-MaLR	0.56
ERV1	1.61
others	0.41

(c) how many single HERV integrations are regulated (exclude multimapping reads and restrict to mappable elements) and how many have ZBTB12 peaks in vicinity?

> When multi-mapping reads were excluded, 9.2% of ERV1 integrations (737/8,028) were differentially expressed after ZBTB12 knockdown (650 up and 87 down). Among them, 81 ERV1 integrations are within 10 kb of ZBTB12 ChIP-seq peaks (74 up and 7 down after ZBTB12 knockdown).

(d) If LTR7 elements are major targets, please show some examples in genome browser screenshots. The example in Fig. 4e needs to be expanded to better identify the transcription unit; also the ZBTB12 peaks appears very close to background based on the scale (in comparison with Fig S10a).

> We noticed that the Y-axis values of Fig. 4e were calculated by different normalization compared to other genome browser screenshots. These are corrected in the revised figure. As suggested, we expanded Fig. 4e and added another example of LTR7/HERVH.

This analysis will help to better judge if ZBTB12 could regulate genes or HERVs or both. In particular HERV regulation appears difficult to follow based on the current analysis. In Figure S5e, the basemean vs log2fc plot for single HERV loci (I assume) show a clear shift towards upregulation, suggesting that almost all HERVs are regulated. This would not correspond with the notion of the authors that only LTR7 HERVs are regulated. There might be a normalization problem here, so it would be important to include elements which are not regulated and which should for a cloud around log2fc of 0.

> The original Figure S5e shows the expression of LTR7/HERVH loci but not all HERVs. To clarify this, we revised the figure, including other HERVs whose expression was not changed by ZBTB12

knockdown (Fig. S5h).

Another possibility is that most of the regulation (HERVs/genes) is explained by the cells assuming a naive-like state upon ZBTB12 KD. In this scenario, ZBTB12 KD triggers this transition, but majority of regulation might be indirect due to the changed transcriptional network.

> Although we identified only 70 ZBTB12 ChIP-seq peaks nearby full-length LTR7/HERVH loci (within 10 kb), ZBTB12 knockdown in hESCs showed global upregulation of almost all LTR7/HERVH loci, which is probably due to both direct and indirect effects of ZBTB12 on LTR7/HERVH regulation. Therefore, we clarified that direct regulation by ZBTB12 is restricted to specific HERVH loci. Please see Page 2 Line 9 and Page 5 Line 3.

Additional comments:

The quality of the NANOG ChIP-seq profile is also questionable, at least the signals shown in Figure S10a/i look very noisy. It is always important to indicate identified peak positions for TFs in the genome browser views. Also it is important to demonstrate pos/neg control regions to judge background/noise.

> As suggested, we used a higher quality NANOG ChIP-seq dataset (GSM2816625) to update the figures (dotted red box in Supplementary Fig. 10a/i). All the data related to NANOG ChIP-seq public data were re-analyzed, and figures were replaced accordingly (Supplementary Fig. 7d, 8d, 9a, 9b, 10a, 10i and Supplementary Table 13). We also added input control for ZBTB12 ChIP-seq to background judgment.

For claiming that there is antagonism between NANOG and ZBTB12 in HERV regulation, HERV target identification of NANOG needs to be more robust.

> We suggested the antagonism based on the opposed results of LTR7/HERVH expression after ZBTB12 or NANOG knockdown (Fig. 4i). However, we agree that the word "antagonism" could be misleading due to the lack of mechanistic interactions between ZBTB12 and NANOG. Thus, we rephrased it in the revised manuscript (Page 16 Line 6, Page 23 Line 2).

The RNA-seq tracks in FigS10a/i also look suspicious. There is very low correlation between the signals and the gene/LincRNA annotation. As these transcripts were used in functional assays it would be important to clarify the exact identity of the regulated transcript (spliced/unspliced/internal promoter used in LINC-ROR?).

> To analyze the expression of retrotransposons and non-coding RNAs, we performed total RNA-seq (not mRNA-seq based on poly-A enrichment) that also captures unspliced transcripts. Thus, the RNA-seq tracks in Fig S10a, i represent both spliced and unspliced transcripts from LINC-ROR and

ESRG loci. Regarding protein-coding genes such POU5F1 below, our RNA-seq showed a clear enrichment of exonic reads.

Reviewer #5 (Remarks to the Author):

For clarity to the authors, I have come into this review process mainly to assess the rebuttal of the points raised by reviewer #3. I have read the manuscript with interest and congratulate the authors on an extensive and well conducted study. I see no major technical issues in this revised version, and can appreciate from the rebuttal that conclusions have been toned down to better reflect what the data show.

> We appreciate the reviewer for finding our work interesting and well-conducted.

The authors have gone through substantial lengths to address the comments from reviewer #3, and the changes made to the manuscript look to be appropriate to me. I only have a relatively minor comment regarding the point on the non-random binding of ZBTB12 to HERVH loci. The reviewer had suggested a comparison to a random control. This seems to have been done for LTRs (Fig. 4b), although methodological details are missing, and these should be added in my opinion. A similar analysis could be done for HERVH specifically, to ask if the observed ZBTB12 peak distribution (overlapped and within 10kb of HERVH) is indeed non-random.

> As suggested, we added the methodological details for the enrichment analysis (Page 37 Line 16). Although we could analyze ZBTB12 enrichment in all LTR loci, a limited number of ZBTB12 peaks which overlap with HERVH loci (3 overlapped peaks) hampered robust statistical analysis.

Reviewer #4 (Remarks to the Author):

The authors have satisfactorily addressed my major concerns.

REVIEWER COMMENTS

Reviewer #4 (Remarks to the Author):

The authors have satisfactorily addressed my major concerns.

> We appreciate this reviewer for constructive comments that improved our work.